# Rhythmicity of neuronal oscillations delineates their cortical and spectral architecture
Vladislav Myrov [1] ✉, Felix Siebenhühner [2,3], Joonas J. Juvonen [1,2], Gabriele Arnulfo [2,4], Satu Palva [2,5] & J. Matias Palva [1,2,5]

Neuronal oscillations are commonly analyzed with power spectral methods that quantify signal amplitude, but not rhythmicity or 'oscillatoriness' per se. Here we introduce a new approach, the phase-autocorrelation function (pACF), for the direct quantification of rhythmicity. We applied pACF to human intracerebral stereoelectroencephalography (SEEG) and magnetoencephalography (MEG) data and uncovered a spectrally and anatomically fine-grained cortical architecture in the rhythmicity of single- and multi-frequency neuronal oscillations. Evidencing the functional significance of rhythmicity, we found it to be a prerequisite for long-range synchronization in resting-state networks and to be dynamically modulated during event-related processing. We also extended the pACF approach to measure 'burstiness' of oscillatory processes and characterized regions with stable and bursty oscillations. These findings show that rhythmicity is double-dissociable from amplitude and constitutes a functionally relevant and dynamic characteristic of neuronal oscillations.

Electrophysiological recordings are characterized by prominent rhythmicity —oscillations—of neuronal activity[1,2]. Neuronal oscillations are fundamental for the temporal coordination of neuronal processing throughout the cortical processing hierarchy and have been widely studied over decades[3–5]. Already the pioneering electroencephalography (EEG) recordings in the 1920s–1930s by Hans Berger[6] *e.e.* observed prominent alpha oscillations and reported their well-established functional consequence what has since been termed the "Berger effect" where alpha power (8–14 Hz) is increased when the eyes are closed and decreased at the opening of eyes. Due to their mechanistic roles in the temporal coordination of neuronal processing, oscillations are involved in a variety of cognitive functions ranging from sensorimotor processes[4,7] and perception[8–10] to higher-level cognitive functions such as attention[11,12], working memory[13,14], cognitive control[15], and long-term memory[16]. Conversely, abnormal neuronal oscillations characterize many neuropsychiatric and neurological disorders[17] such as atypical development[18], depression[19], Parkinson's disease[20], schizophrenia[21], and epilepsy[22].

Different features of spontaneous neuronal oscillations have been characterized using resting-state data where event-related activities do not confound the analysis[23–27]. Spectral analysis has established that spontaneous oscillations are similarly present in magnetoencephalography (MEG) and stereo-EEG (SEEG)[28,29], show frequency-specific correlations with fMRI BOLD signal[24] and that power-spectral-peak frequencies of neuronal oscillations exhibit both a medial-to-lateral and a posterior-to-anterior hierarchy gradient of increasing frequency[30].

Yet, after decades of research in various fields, data analysis methods for neuronal oscillations still remain under active development[31–33]. Oscillations are traditionally operationalized by their amplitude and widely studied using power spectral methods, where a spectral peak above the "$\frac{1}{f}$" aperiodic component constitutes "gold standard"[2] evidence for the presence of oscillations, i.e., for significant rhythmicity. In this approach, oscillations are thus defined to be present at frequencies that exhibit greater power than the aperiodic component, which can then be quantified with analyses based on parameterization of power spectra[34,35].

However, being amplitude-based, this approach is only a qualitative indicator of rhythmicity and does not measure it directly nor quantify it explicitly. For example, a peak in the power spectrum shows that oscillations exist in the data, but cannot indicate exactly how rhythmic they are, as the power spectral density (PSD) peak magnitudes are dependent on the signal

[1]Department of Neuroscience and Biomedical Engineering, Aalto University, Espoo, Finland. [2]Neuroscience Center, Helsinki Institute of Life Science, University of Helsinki, Helsinki, Finland. [3]BioMag Laboratory, HUS Medical Imaging Center, Helsinki, Finland. [4]Department of Informatics, Bioengineering, Robotics and System Engineering, University of Genoa, Genoa, Italy. [5]Centre for Cognitive Neuroimaging, School of Psychology and Neuroscience, University of Glasgow, Glasgow, UK. ✉e-mail: vladislav.myrov@aalto.fi

per se. Further, the lifetimes of oscillations cannot be unambiguously deduced from the PSD peak.

Even though the width of the PSD peak is dependent on rhythmicity and can be quantified, several confounders limit the interpretability and preclude the conclusion that a change in said width could be quantitatively associated with a change in rhythmicity. This is because rhythmicity is by definition a property contained in the temporal stability of the phase of a complex signal, and it is independent of its amplitude (Fig. 1e, j).

Next, depending on the nature of the underlying dynamics, neuronal oscillations exhibit a wide range of amplitude distributions[36] ranging from Gaussian, to non-normal, to bistable[37], which is evidenced by their heavy-tailed (gamma-like) amplitude distributions[38] rather than them exhibiting Rayleigh distributed amplitudes expected for Gaussian processes.

Oscillation amplitude envelopes also exhibit power-law long-range temporal correlations in the neural data[39] and computational models[40], which is incompatible with the notion of them being Gaussian. All of those caveats make it impossible to describe oscillations using only power spectrum.

In addition, the neural activity is organized in evolving patterns in different timescales across multiple species[41]. In case of the brain oscillations, such time scales could be operationalized with cycles of different frequencies aka basic units of oscillations. There were attempts in previous studies to take such "brain clock" into account[42] but is still operates with a signal power and requires manual selection of a central frequency which may be different between conditions or subjects[43].

Here we advance a new approach and operationalize the construct of rhythmicity with the phase Autocorrelation Function (pACF). It specifically

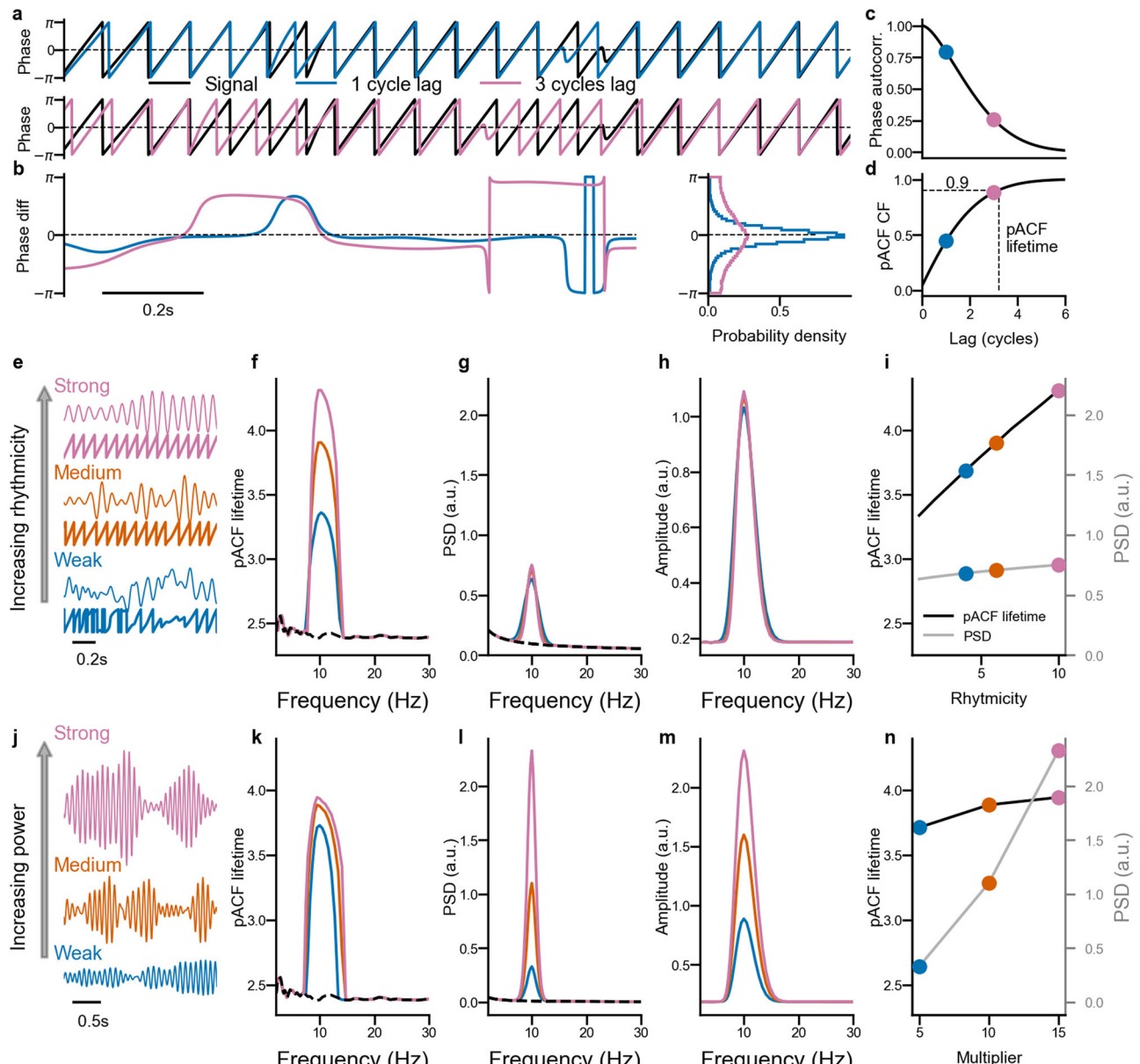

**Fig. 1 | Phase autocorrelations index rhythmicity—validation with simulated data. a** Phase time series of a simulated analytical signal and the same signal at lags of one and three cycles. **b** Time series and histograms of the phase difference between the signal and 1-cycle and 3-cycles lagged version. **c** Phase autocorrelation, obtained as the absolute value of the complex average of the phase differences, as a function of lag. **d** Cumulative Function (CF) of the corresponding pACF. The dashed line indicates the 90th percentile threshold, which is used to determine the lifetime of the oscillation. **e** Example signals and its phase timeseries with strong, medium, and weak rhythmicity, and their (**f**) pACF lifetime, (**g**) power spectral density (PSD), and (**h**) wavelet amplitude spectra (colors as in **e**). **i** pACF lifetime (black) and PSD (gray) of the oscillation peak frequency as a function of rhythmicity. **j** Example signals with Strong, Medium, and Weak power, and their (**k**) pACF lifetime, (**l**) PSD, and the (**m**) wavelet amplitude spectra (colors as in **k**). **n** pACF lifetime (black) and PSD (gray) of the oscillation peak frequency as a function of signal power.

quantifies the rhythmicity of neuronal oscillations in an amplitude-independent manner, operates in the oscillatory cycles time domain and expresses the predictability of a future phase as a function of time (lag), thereby yielding a direct measurement of temporal stability, i.e., rhythmicity.

## Results

### Operationalization of rhythmicity with phase autocorrelations

The first step in the quantification of phase autocorrelations and estimation of the pACF is to acquire the phase time series using complex wavelet filtering (Fig. 1a). The phase difference between this time series and its time-lagged copies (Fig. 1b) is systematic at short lags and becomes increasingly erratic for longer lags, which reflects the decay of temporal (phase) autocorrelations with increasing lags. This intuitive finding is quantified in the pACF by expressing a phase-autocorrelation value for each lag (Fig. 1c, for a detailed description see "Methods").

The pACF can then be aggregated into a single value to indicate the "lifetime" of phase autocorrelations similar to how exponential decay is indexed by a decay time constant. However, as we make no assumptions about the shape of the pACF curve, we use a model-free approach by testing where the CDF crosses a threshold. The lifetime is estimated in a non-parametric manner by transforming the pACF into the corresponding cumulative function and detecting the first lag that exceeds a pre-defined threshold (Fig. 1d, see "Methods" for details). We used here a threshold of 0.9 that thus indicates that 90% of the total observed above-chance-level phase autocorrelation is accounted for by the corresponding lag.

To validate that pACF can be used to quantify rhythmicity, we generated signals with varying phase autocorrelations and power. First, increasing the rhythmicity of the signal (Fig. 1e) resulted in increased pACF lifetimes (Fig. 1f, i) while the power (Fig. 1g, i) and wavelet amplitude spectra (Fig. 1h) remained essentially unchanged. On the other hand, increasing the power of the signal (Fig. 1j) led to salient changes in the power (Fig. 1l, n) and amplitude (Fig. 1m) spectra, while the pACF lifetimes remained (Fig. 1k, n) nearly constant. These results thus demonstrate that pACF taps into "oscillatoriness", i.e., stability of the oscillation while amplitude-based metrics tap into the strength of the oscillating processes, and these two approaches may thus uncover distinct mechanistic or functional characteristics of oscillations in vivo.

### Phase autocorrelations reveal sparse oscillations in mesoscale human cortical assemblies

To assess the presence of phase autocorrelations in human cortical oscillations, we first analyzed intra-cerebral stereo-EEG (SEEG) resting-state recordings. SEEG electrode contacts in cortical gray matter yield direct recordings of the local field potentials generated by mesoscale neuronal populations. First, in a single subject, an inspection of the low-alpha frequency band (7.9 Hz) pACFs (Fig. 2a) showed that some contacts exhibited strong phase autocorrelations compared to the null level.

To test whether the observed pACF lifetimes were significantly above the noise level, the significance threshold corresponding to $p \leq 0.01$ was identified as the 99th percentile of the surrogate pACF lifetimes (Fig. 2b, see "Methods"). This showed that while distribution of pACF lifetime values for contacts without oscillations was similar to the distribution of noise-level values (Supplementary Fig. 1d) a subset of brain areas exhibited significant oscillations (Fig. 2c).

We then extended this approach to all frequencies for 81 wavelet frequencies from 2 to 100 Hz with logarithmic spacing to obtain the spectra of phase autocorrelation lifetimes, hereafter called "pACF spectra", and used hierarchical clustering to group the electrode contacts with similar pACF spectra. Spectral clusters of oscillations were predominantly found in alpha and beta frequency bands with peaks at 7.8 Hz and 22.9 Hz (Fig. 2d). Strikingly, the clusters in both bands were anatomically well segregated and spectrally composed of very few narrow-band components.

To compare these findings with power spectral analysis, we evaluated PSD spectra for these data using the separation of the spectra into aperiodic

"$\frac{1}{f}$" and periodic components (Fig. 2e) with the FOOOF method[35] (see "Methods"). Both the pACF (Fig. 2f) and PSD spectra exhibited main peaks at the same alpha and beta frequencies, indicating converging inference of oscillations therein. However, neither the absolute PSD peak values nor their height above the $\frac{1}{f}$ fit was correlated with pACF-based rhythmicity of the oscillations (Fig. 2g, h, Pearson correlation coefficient 0.06 for the peak at 7.8 Hz and $-0.001$ for the peak at 22.9 Hz). In addition, while pACF identified significant rhythmic oscillations in narrow peaks at 7.8 Hz, 9 Hz and 22.9 Hz of only 2–3 Hz width, the corresponding spectral peaks in PSD were 64% wider, and the spectral power was elevated above the $\frac{1}{f}$ levels also between the peaks and thus outside of the frequencies where pACF revealed significant rhythmicity (see Fig. 2f dashed line). Moreover, clustering of the electrode contacts with FOOOF'ed PSD failed to uncover the four clusters (Supplementary Fig. 2a, b) revealed by pACF-lifetime-based clustering (Supplementary Fig. 2c). In line with the simulations (Fig. 1i, n), these findings thus suggest that rhythmicity is a unique property of neuronal oscillations in vivo and complementary to their power.

### Rhythmicity delineates neuronal oscillations in highly specific anatomical and spectral domains

To assess how rhythmicity, in contrast with amplitude, defines the cortical architecture and spectral structure of neuronal oscillations, we computed phase autocorrelations and power spectra at the population level from 10-min resting-state SEEG ($N = 64$ subjects) and source-modeled MEG data ($N = 54$ subjects, $N = 204$ recordings). We represented the group-level data in the Schaefer atlas[44] at the resolution of 400 parcels and first obtained grand-average pACF and PSD spectra for each parcel (Fig. 3a). Both SEEG and MEG data showed that rhythmicity in the human brain was largely contained in the alpha (5–11 Hz in SEEG and 7–14 Hz in MEG) and beta (12–32 Hz in SEEG and 15–32 Hz in MEG) frequency bands. Quantification of the extent of cortical areas exhibiting significant rhythmicity showed that 29% of electrodes in individual subjects in SEEG and 42% of parcels in individual MEG subjects exhibited at least one significant ($p < 0.01$, as in Fig. 2b) pACF lifetime value in the theta-alpha (5–15 Hz) frequency bands. Activity in the beta peak was significant in 18% of each of the electrodes in SEEG and 18% in parcels in MEG (Fig. 3b). These findings illustrate that neocortical oscillations may not be as ubiquitous as thought earlier. Interestingly, pACF lifetime indicated more oscillatory activity in the alpha than in the beta band, while PSD indicated more significant activity in the beta than in the alpha band (Supplementary Fig. 3a).

To identify the anatomical sources of these oscillations, we first estimated the similarity of anatomical patterns of pACFs between different frequencies. The source anatomies in SEEG and MEG were split into six distinct clusters of frequencies (Fig. 3c, see Methods for details). Central frequencies localized the rhythmicity in SEEG into prefrontal theta (5.6 Hz, Fig. 3d), posterior low-alpha (7.1 Hz, Fig. 3d), a mixture of visual and sensorimotor high-alpha (9.1 Hz, Fig. 3d), sensorimotor and attentional low (16.9 Hz, Fig. 3d) and mid (25.5 Hz, Fig. 3d) beta, and attentional and frontoparietal high-beta (31.6 Hz, Fig. 3d). Except for the prefrontal theta, these were well co-localized with the canonical tau-, alpha-, and mu rhythm-like components in MEG[45]: superior temporal low-alpha (8.3 Hz), posterior alpha (10.1 Hz), and sensorimotor high-alpha and beta (14.6 and 26.0 Hz), respectively (Fig. 3e). Such fine-grained frequency communities were, however, less salient in power spectral analysis that also was not able to separate low and high-alpha clusters (Supplementary Fig. 3b) and yielded more anatomical spread than the pACF lifetimes (Supplementary Fig. 3d–i). These findings demonstrate that rhythmicity uncovers spatial and spectral characteristics of neuronal oscillations that are distinct from those observable with their power spectra.

### Distinct functional neuroanatomy for single- and multi-band mesoscale oscillations

Visual analysis of individual subjects (Fig. 2) in SEEG suggested that in the mesoscale neuronal populations, genuine oscillations are spectrally sparse

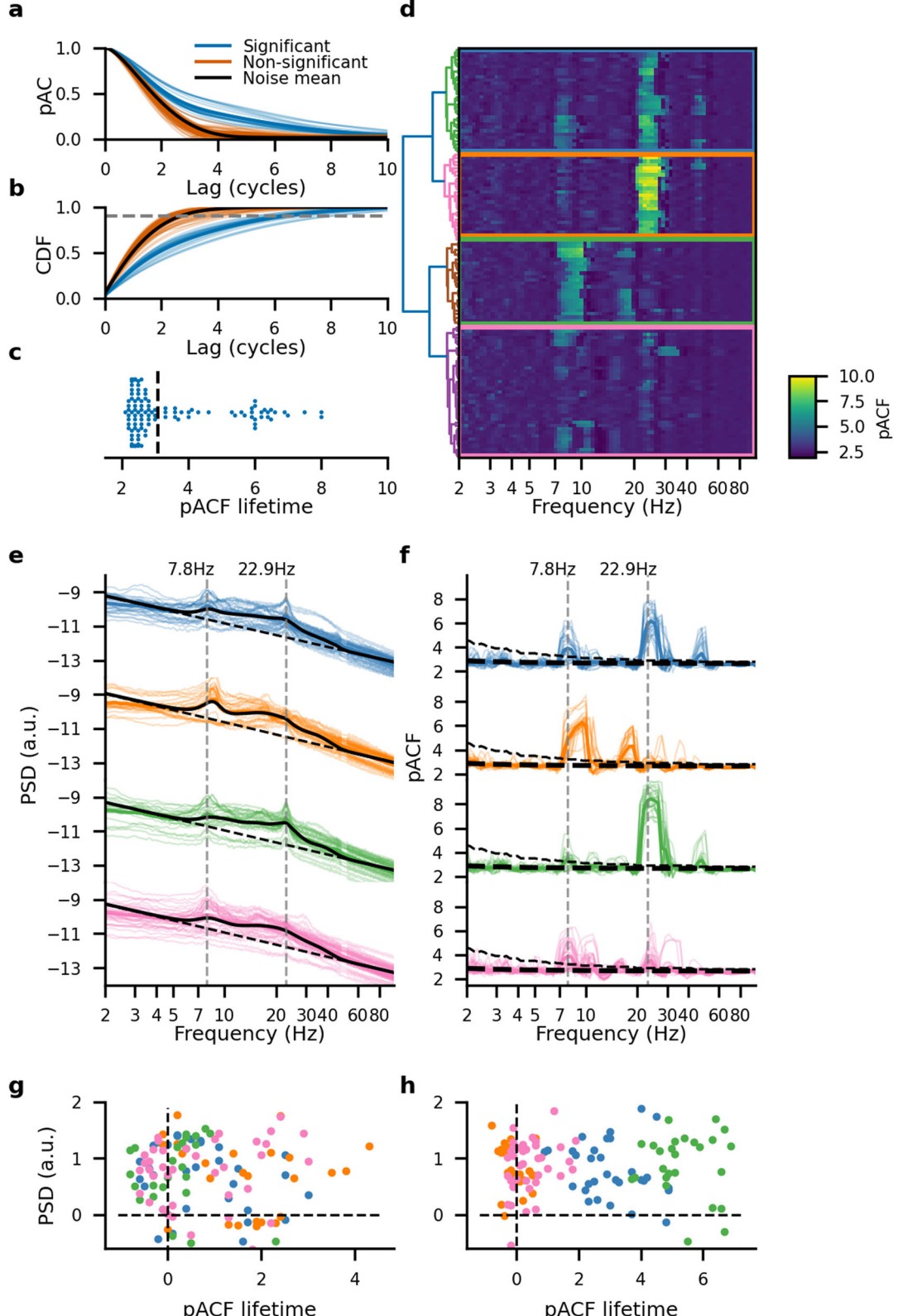

**Fig. 2 | Phase autocorrelation profiles are separated in well-delineated frequency clusters. a** The pACF and (**b**) its cumulative function (CF) for each gray matter SEEG electrode contact for a representative subject, thin lines indicate individual contacts while thick lines indicate a mean value. **c** swarm plot of pACF lifetime at a single frequency of 9.5 Hz. The dashed line indicates the lifetime threshold. **d** Individual electrodes' pACF spectra. The pACF spectra were sorted by hierarchical clustering using Ward's criterion and partitioned into four clusters (colored squares corresponding to the dendrogram). **e** Power Spectrum Density (PSD) for each cluster. The thick black line indicates an average spectrum, the dashed line indicates an average aperiodic component and the dotted gray line indicates the peak frequencies of the subject (7.8 Hz and 22.9 Hz). **f** pACF lifetime spectra for each cluster. The thick black dashed line indicates the mean pink-noise pACF lifetime, dashed black line indicate the surrogate level (99 percentile of filtered pink-noise pACF lifetime), and dotted gray lines indicate the peak pACF frequencies of the subject (7.8 Hz and 22.9 Hz). **g, h** Difference between PSD peak value with the aperiodic fit and pACF peak value with noise level for two peak frequencies of 7.8 Hz (**g**) and 22.9 Hz (**h**).

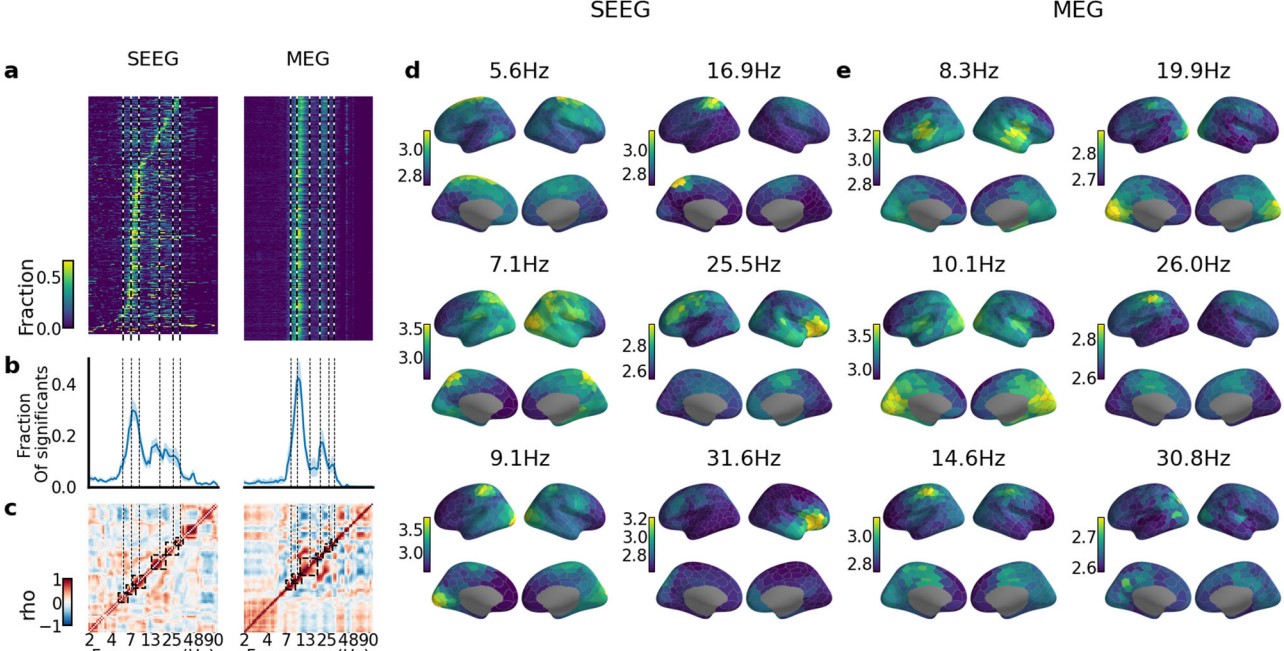

**Fig. 3 | Phase-autocorrelation similarity revealed nested frequency clusters.**
**a** Fraction of significant pACF lifetime spectra for each parcel sorted by peak frequency and averaged across subjects for SEEG (left) and MEG (right). **b** Spectra for fractions of cortical parcels exhibiting significant rhythmicity in pACF (thick line: cohort average, shadowed areas indicate confidence intervals estimated via bootstrapping, $N$ bootstraps = 1000, 5th and 95th percentiles were used). **c** Correlation of the anatomical patterns of pACF lifetime for each pair of frequencies (rectangles indicate frequency clusters). Anatomy of the average pACF lifetime across frequencies in each cluster for SEEG (**d**) and MEG (**e**) datasets (see **c**).

and observed only in one or a few frequency bands. To quantify this at the population level, we assessed the fraction of SEEG electrode contacts or MEG parcels in functional brain systems (Fig. 4a) exhibiting at least one significant peak in the pACF lifetime spectra at any frequency. We found that the dorsal attention, visual, and sensorimotor systems exhibited the highest prevalence of oscillations (70–80%, Fig. 4b), while the default (62%, 59%, and 57% for DefaultB, DefaultC, and DefaultA systems) and limbic (60% and 39% for LimbicA and LimbicB systems) systems had the smallest fractions of oscillations.

Anatomically, oscillations were the scarcest in the anterior frontal cortex (Fig. 4c). While an average of 66% of SEEG electrode contacts and 73% of parcels in MEG exhibited oscillations as evaluated with pACF, only 27% of contacts exhibited two distinct peaks (29% for MEG), and 11% (7% for MEG) more than two peaks in any frequency band (Fig. 4d). At the level of brain systems, the sensorimotor system exhibited the greatest fraction of individual electrodes or parcels oscillating concurrently at 2 (35% for both SEEG and MEG) or at 3 or more bands (19% for SEEG and 11% for MEG). Neuronal oscillations with true, above-chance level rhythmicity are thus sparse in the frequency domain and found largely in only one or two bands in both SEEG and MEG even though the power-spectral peaks extended over greater frequency ranges.

The single-band oscillations in SEEG were predominantly localized to occipital, temporal, and posterior parietal cortices (Fig. 4e), while the multi-band oscillations (comprised of multiple frequencies) were the most prevalent in central frontoparietal areas, especially in the sensorimotor and dorsal attention systems (Fig. 4f). We reproduced this analysis with MEG data and found that the anatomical distributions of single- and dual-band oscillations therein were similar to those found in SEEG (Fig. 4k, l), which corroborated that this division is not limited to mesoscale neuronal assemblies but rather reflects a systematic functional division in the architecture of neocortical oscillations. We then further assessed the anatomical distributions of peak frequencies for the single- and multi-band oscillations.

Alpha-band components of multi-band oscillations exhibited significantly higher peak frequencies than single-band alpha oscillations in both SEEG and MEG (Fig. 4g, m, $p = 0.0005$ for the SEEG cohort, $p = 0.0001$ for the MEG cohort, shuffle test). This division was salient in the topographies of alpha peak frequencies (Fig. 4h, i, n, o), which revealed bilaterally symmetric peak frequency subdivisions across the neocortical surface. Importantly, the beta oscillations were much more likely to appear as a component in multi-band (26% of SEEG electrodes and 38% of MEG parcels) oscillations than alone as single-band oscillations (9% of SEEG electrodes and 6% of MEG parcels).

## Phase autocorrelations and power spectra tap into distinct underlying constructs in oscillatory brain dynamics

In order to assess the relationship between spectral power and rhythmicity quantitatively, we tested how the cortical topographies of pACF, PSD, and wavelet amplitudes were correlated. We first assessed whether the wavelet amplitudes or pACFs were more predictive of the PSD observations by evaluating partial correlations among these metrics for the SEEG and MEG datasets. The correlations between PSD and amplitude were stronger than those between PSD and pACF essentially throughout the analyzed frequency spectrum (Fig. 5a, b) and prominent in the alpha (7.8 Hz for SEEG data, 10.5 Hz for MEG data) and beta (14.8 Hz for SEEG data, 20.8 Hz for MEG data) frequencies (Fig. 5c, d).

Already the pioneering electroencephalography (EEG) recordings in the '50s by Hans Berger[6] *e.e.* observed prominent alpha oscillations and their well-established functional feature—the "Berger effect"—where alpha power is increased when the eyes are closed and decreased at the opening of eyes. To validate this result further, we repeated this analysis with independent eyes-closed and eyes-open resting-state MEG data and found that they followed the same pattern (Supplementary Fig. 5a, b). This shows that the resting-state power spectra are more determined by the amplitude of neuronal oscillations than by their rhythmicity. We then assessed whether the "Berger effect" was driven by a change in amplitude or rhythmicity.

Interestingly, while pACF, PSD, and amplitudes were all greater for the eyes closed than for the eyes-open condition, the effect size of the Berger effect was greater for pACF than for PSD or amplitude (Cohen's $d' = 1.59$ for

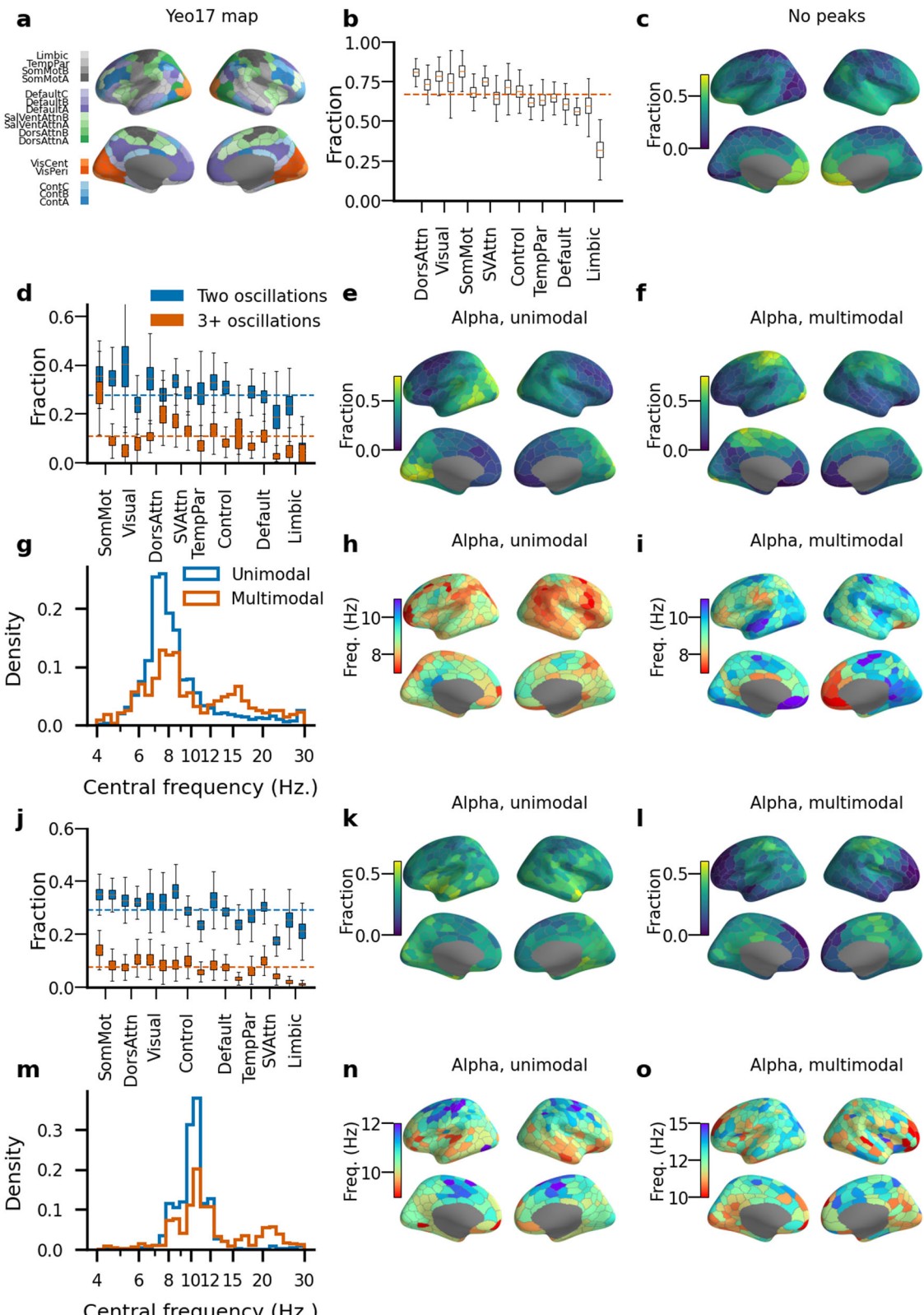

**Fig. 4 | Architecture of single and multi-frequency oscillations. a** System assignment ($N = 17$) for each of 400 parcels in the Schaefer parcellation. **b** Fraction of contacts in individuals with at least one significant oscillatory peak aggregated across systems and sorted by the mean value of corresponding Yeo-17 system. **c** Fraction of individual contacts without any significant oscillations in the SEEG dataset. Fraction of individual contacts (SEEG, **d**) and cortical regions (MEG, **j**) with two or more than three oscillation components. Fraction of individual contacts (SEEG, **e**) and cortical regions (MEG, **k**) with a single oscillation in the alpha band (SEEG **e**, MEG **k**) or with multiple oscillatory components including one in the alpha band (SEEG **f**, MEG **l**). Histogram of peak frequencies pooled across individual contacts (SEEG, **g**)/cortical regions (MEG, **m**) for signals with single or multiple peaks. The peak frequency of a parcel analyzing single-peak contacts and contacts with several peaks for the alpha band (all three sub-bands combined, SEEG **h**, **i**, MEG **n**, **o**). The box ends in (**b**, **d**, **j**) indicate the lower quartile (Q1, 25th percentile) and upper quartile (Q3, 75th percentile) across bootstrapped fraction ($N$ bootstraps = 1000), notches indicate the median, and whiskers indicate the range of Q1 − 1.5 * IQR and Q3 + 1.5 * IQR, where IQR is the inter-quartile range (Q3 − Q1).

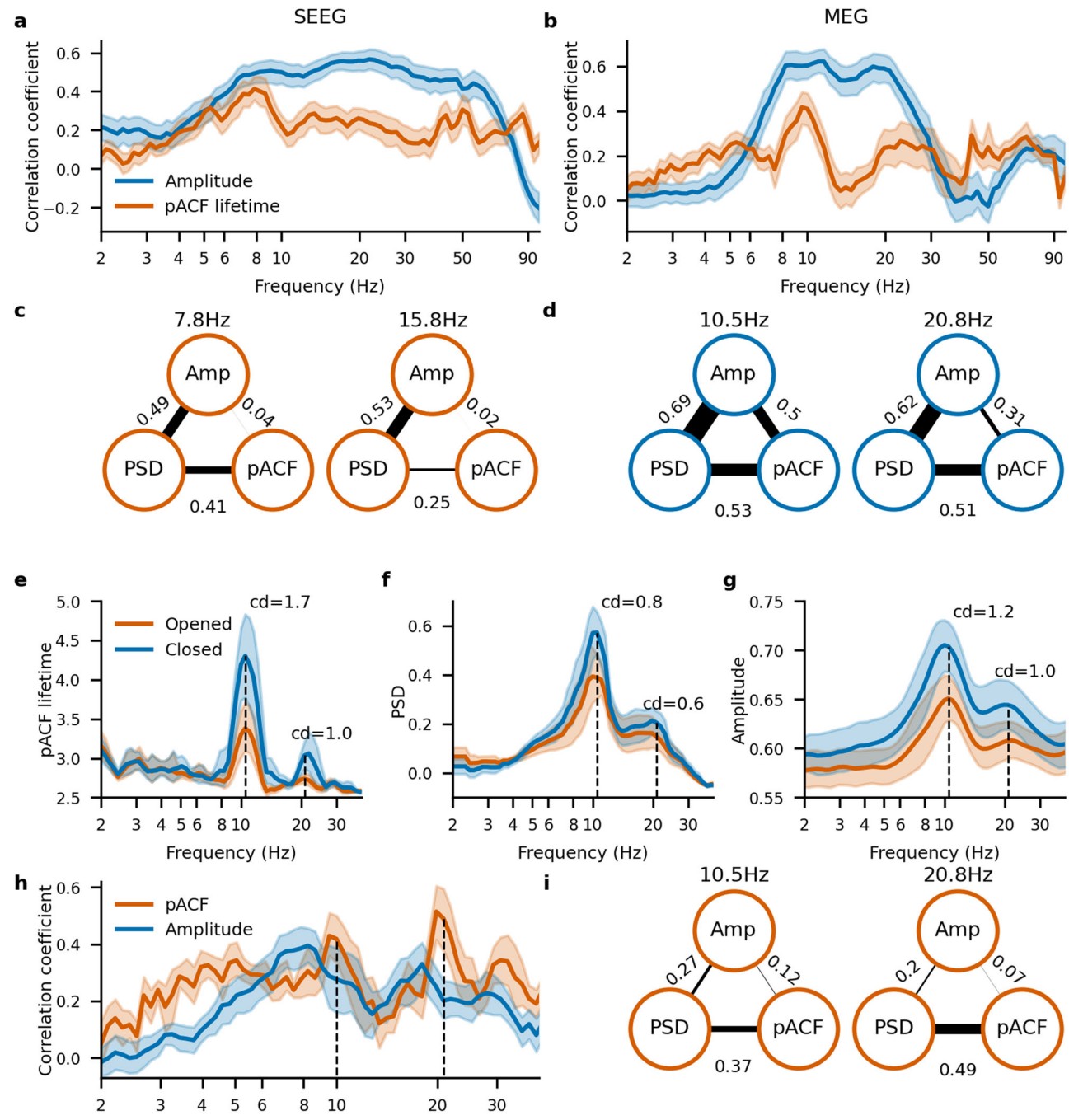

**Fig. 5 | Phase autocorrelation and power spectra reflect different mechanisms.** Pearson correlation coefficient of PSD with amplitude (blue) and pACF (orange) lifetime values as a function of frequency for the SEEG (**a**) and MEG (**b**) datasets. Pearson correlation triangle for each combination of PSD, amplitude, and pACF lifetime at peak alpha (7.8 Hz for the SEEG and 10.5 Hz for the MEG datasets, left) and beta (15.8 Hz for the SEEG and 20.8 Hz for the MEG dataset, right) frequencies for the SEEG (**c**) and MEG (**d**) datasets. pACF lifetime (**e**), PSD (**f**), and amplitude spectra (**g**) for the resting state recordings with opened and closed eyes obtained via MEG recordings. **h** Pearson correlation coefficient of contrast values between eyes open and closed conditions for PSD with amplitude (blue) and pACF lifetime (orange) as a function of frequency. **i** Pearson correlation triangle for each combination of PSD, amplitude, and pACF lifetime at peak alpha (10.5 Hz, left) and beta (20.8 Hz, right) frequencies. Shaded areas indicate bootstrapped confidence intervals (5th and 95th percentiles, N bootstraps = 1000).

pACF lifetime, 0.76 for PSD, and 1.05 for amplitude, calculated for the peak frequencies). Partial correlation analysis showed that, in contrast with PSD being driven by amplitude in the resting state, the eyes-closed-vs-open difference in the PSD spectra was more strongly predicted by pACF in alpha and beta frequencies (0.42 and 0.49) than by amplitude (0.28 and 0.20) (Fig. 5h, i). This indicates that the long-known Berger effect in PSD in fact reflects more a change in alpha rhythmicity than in its amplitude and thus

that rhythmicity is a functionally significant characteristic of neuronal oscillations.

### Rhythmicity is a prerequisite for inter-areal synchronization of neuronal oscillations

Long-range phase synchronization is fundamental for neuronal communication in brain networks[46] but its dependence on the rhythmicity of

the coupled oscillating neuronal assemblies has remained unaddressed. To answer this question, we compared pACF with long-range synchronization first with computational modeling and then with SEEG and MEG data. Using a new hierarchical Kuramoto model (see "Methods"), we simulated cortical synchronization dynamics emerging in the human structural connectome. The model was set to operate in a regime of realistic local (pACs) and global (inter-areal synchronization) dynamics. Inter-areal synchronization was measured with the Phase-Locking Value

(PLV) in the model and SEEG data and with the weighted Phase Lag Index (wPLI) in MEG data.

We first estimated the correlation between pACF and node strength, i.e., the mean synchronization of a network node (model), a contact (SEEG), or a parcel (MEG) with all other nodes/contacts/parcels. We found that in the model, node strengths were positively correlated with pACF around the central frequency of the model (Fig. 6a, b). Testing this prediction with empirical data, we found that, indeed, synchronization and pACFs exhibited

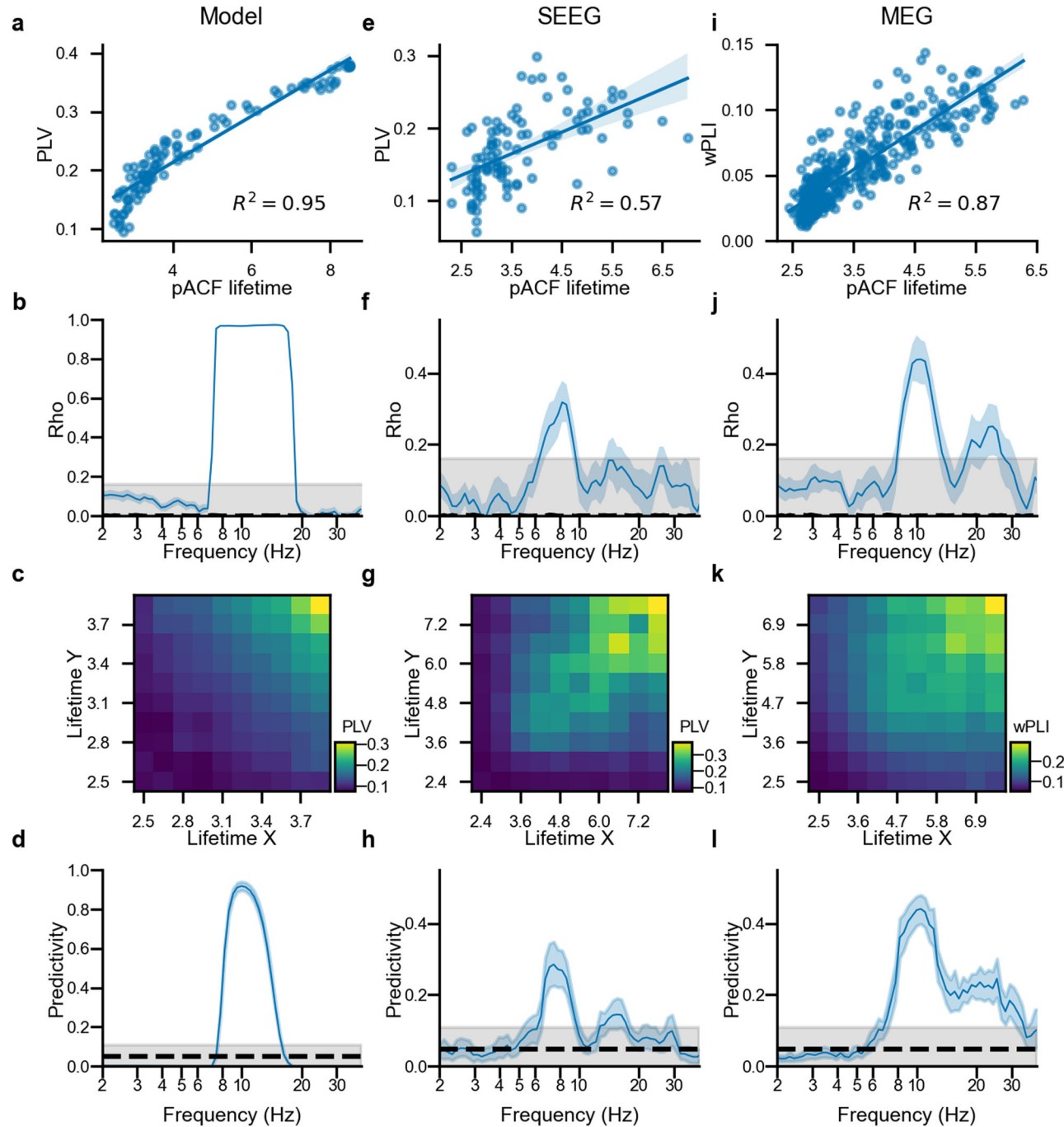

**Fig. 6 | Phase-autocorrelation as pre-requisite for phase synchrony. a, e, i** Average PLV (model and SEEG) or wPLI (MEG) for each node as a function of corresponding pACF lifetime for single model realizations or an individual subject (SEEG and MEG) at the alpha frequency (10 Hz for the model, 8.6 Hz for SEEG data and 11 Hz for the MEG data). **b, f, j** $R^2$ for a fitted linear regression as a function of frequency. The shaded areas indicates confidence intervals estimated with bootstrapping (*N* rounds = 1000, 5th and 95th percentiles). The gray area indicates the

null-level estimated by a random permutation of the node strength vector (*N* permutations = 1000). **c, g, k** Phase synchrony in the alpha frequency (10 Hz for the model, 8.6 Hz for SEEG data, and 11 Hz for the MEG data) binned by pACF lifetime of two contacts (SEEG) or parcels (MEG). **d, h, l** Predictability of significant synchrony between two nodes (model), contacts (SEEG), or parcels (MEG) as a function of frequency. The gray area indicates the null-level estimated by a random permutation of the 2D phase synchrony heatmap (*N* permutations = 1000).

significant positive correlations in both SEEG and MEG in the alpha frequency band (6.5–9.5 Hz, SEEG; 8–13 Hz, MEG Fig. 6e, f), while MEG also revealed positive correlations in the beta (18–26 Hz) band (Fig. 6i, j). Thus, in both model and brain, nodes with high rhythmicity also exhibited greater synchronization.

We next hypothesized that rhythmicity would, in fact, be a mechanistic prerequisite for phase synchrony between two neuronal populations. We first addressed this qualitatively and sorted samples according to the pACF lifetimes of each pair of nodes (X and Y) and evaluated the phase synchrony between these nodes in pACF lifetime bins (Fig. 6c, g, k). This approach showed that both in the model (Fig. 6c) as well as in SEEG and MEG (Fig. 6g, k), strong synchronization was found exclusively when both nodes exhibited high rhythmicity.

To further quantify this, we evaluated the predictability of significant synchrony between two nodes (3.42*surrogate mean, corresponding to $p < 0.001$, Rayleigh distribution) if both of them have significant rhythmicity (95th percentile of noise pACF lifetime was taken as a threshold, similarly to the previous analysis). Predictability was defined as $\frac{tp}{tp+fn}$ where $tp$ is the number of true positives (edges with both significant synchrony and pACF lifetime in both nodes) and $fn$ the number of false negatives (edges with significant synchrony but at least one node does not show significant pACF lifetime).

In the model, predictability higher than the surrogate level was found around the central model frequency (Fig. 6d). Importantly, in both SEEG and MEG, we observed significant predictability in the alpha (6.5–9.5 Hz for SEEG, 8–13 Hz for MEG) and beta frequency bands (13–17 Hz for SEEG, 18–24 Hz for MEG, Fig. 6h, l). Thus, robust neuronal synchronization in the human brain is predominantly found only among cortical areas that exhibit strong rhythmicity.

## The decay shape of phase autocorrelations dissociates stable and bursty oscillations
Both animal-model[47] and human electrophysiological studies[48] suggest that the rhythmicity of neuronal oscillations may vary qualitatively from continuous "meta-stable" oscillations to apparently discrete bursts lasting only a few cycles. Brief high-amplitude bursts[49,50] have been shown to be a robust neuronal phenomenon and to play a role in cognitive functions[51], but can not be dissociated from sustained and temporally stable oscillations with power-spectral amplitude-based methods.

Bursts are typically assessed as periods with amplitude exceeding a threshold in their peri-event time but there is ongoing debate about how to characterize the burstiness of a signal as a whole. Here, we introduce a pACF- and phase-based approach to operationalize the "burstiness" or "oscillatoriness" of the signal with the "stability index". We defined "bursts" as short periods of oscillations that are un-phase-correlated with their peri-burst time series and "stable oscillations" as a process that exhibits long-range phase autocorrelations evidenced by heavy-tailed pACFs (Fig. 2a).

At first, we validated the method using simulated signals with varied inter-burst intervals (mean value from 5 to 30 cycles, standard deviation of 0.25 cycles) and burst length (mean value from 3 to 20 cycles). The stability index correctly characterized as stable (stability index > 0.05) time series with long periods of stable oscillations (and short inter-burst intervals), and signals composed of short bursts at large intervals as bursty (stability index < −0.05) (Fig. 7d).

We then estimated stability index spectra for the SEEG and MEG cohorts using only those parcels, electrode contacts, and frequency bands where significant rhythmicity was observed in primary pACF analyses (Fig. 3a, b). We found that the theta and alpha bands (6–11 Hz for SEEG, 7–15 Hz for MEG) exhibited both the greatest average stability index values (Fig. 7e) and the largest fractions of oscillations classified as stable (Fig. 7f, g).

Analyzing SEEG data on the system level, we found that the Limbic system had the highest fraction of stable oscillations in the alpha frequency band but also the highest fraction of burst-like activity in the beta band. In the MEG data, stable oscillations dominated in all frequency clusters and all subsystems. However, the beta-band cluster showed a higher number of

burst-like activity in comparison to the alpha band. These data are in line with recent findings from an animal in vivo electrophysiology[47].

## Rhythmicity is dynamically modulated in event-related processing underlying stimulus detection
Finally, we aimed to assess whether rhythmicity is a stationary property of local neuronal oscillations or modulated dynamically in response to event-related task demands. We evaluated time-resolved pACF in MEG data acquired during a visual threshold-stimulus-detection task (TSDT)[8–10]. TSDT is a continuous performance task where sensory awareness is probed by constant-intensity stimuli calibrated to the threshold of detection (50 % detection (hit) rate (HR), final mean HR = 41%). Hence, TSDT is associated with minimal sensory stimulation and as such is well-suited for probing the changes in spontaneous brain dynamics imposed by task demands[52]. We first reproduced the classical TSDT observations: as observed earlier[8–10], the detected stimuli (Hits) were characterized by much greater stimulus-phase-locking and induced amplitude responses than the undetected stimuli (Misses, Supplementary Fig. 6a).

To avoid possible bias in the Hit vs. Miss contrast due to a varying number of trials, the average Hit/Miss response was bootstrapped using the minimum number of trials. The time-frequency representations (TFRs) showed that also the pACF exhibited event-related responses (Supplementary Fig. 6b). The differences in pACF between the Hit and Miss conditions were observed in four time-frequency windows (Fig. 8a) with up to 39% of cortical parcels exhibiting significant Hit-Miss differences (Fig. 8b, permutation test, $N$ permutations = 1000, 95th percentile of surrogate maximum difference across time). In pACF, the event-related response was characterized by a decrease in stability in low-alpha frequency (7.5–10 Hz) in 27% of recordings, primarily in the control and attention systems (Supplementary Fig. 6c, d) but also by a decrease in stability in the beta frequency (20–27 Hz) localized primarily to the somatomotor system (Fig. 8c, f).

Comparing the pACF results with the previously established data of induced oscillatory response (Supplementary Fig. 6e, f) and Phase-Locking Factor (PLF, Supplementary Fig. 6i, j), we found that pACF yielded partially distinct responses in both dynamics and peak frequencies from those of stimulus locking and induced responses. pACF revealed reduced rhythmicity in alpha and beta bands in the 0.25–0.75 s time window similar to reduced induced oscillatory responses in alpha and beta bands (Supplementary Fig. 6g, h) indicating that there was indeed a genuine change in rhythmicity in these frequencies.

In summary, pACF revealed a fine-grained frequency resolution in comparison to the induced response (for instance, the alpha and beta effects showed as a single cluster using the induced response approach but were separate when using pACF lifetime, Supplementary Fig. 6g, h). pACF thus yields novel insight into the event-related dynamics by enabling the dissociation of the early non-oscillatory broadband responses from the genuine changes in oscillatoriness of spontaneous oscillations.

## Discussion
Neuronal oscillations encompass a wide range of neuronal processes where the phase of the oscillation is the key mechanistically and functionally substantial element[25,53,54], they predict visual perception[55], plays a role in neural coordination[56], plasticity[57] and reflect phase-specific functional relations between neuronal populations[58]. Phase-synchronization is also a basis for resting-state networks[27,59,60]. The relationship between phase and amplitude correlations[23,61] and that between oscillations of different frequencies[28] have remained an active topic for investigation.

Oscillations have variable rhythmicity, ranging from stable "classical" oscillations such as the alpha rhythm[62–64] to short-lived bursts[65], and to quasi-periodic oscillations that are genuine oscillations but have such variability in frequency that in time-averaged metrics such as the PSD, they average out to an apparent "$\frac{1}{f}$"-like power spectrum[4,66]. However, while the amplitude dynamics and inter-areal correlations of neuronal oscillations are widely studied, rhythmicity per se has attracted little attention.

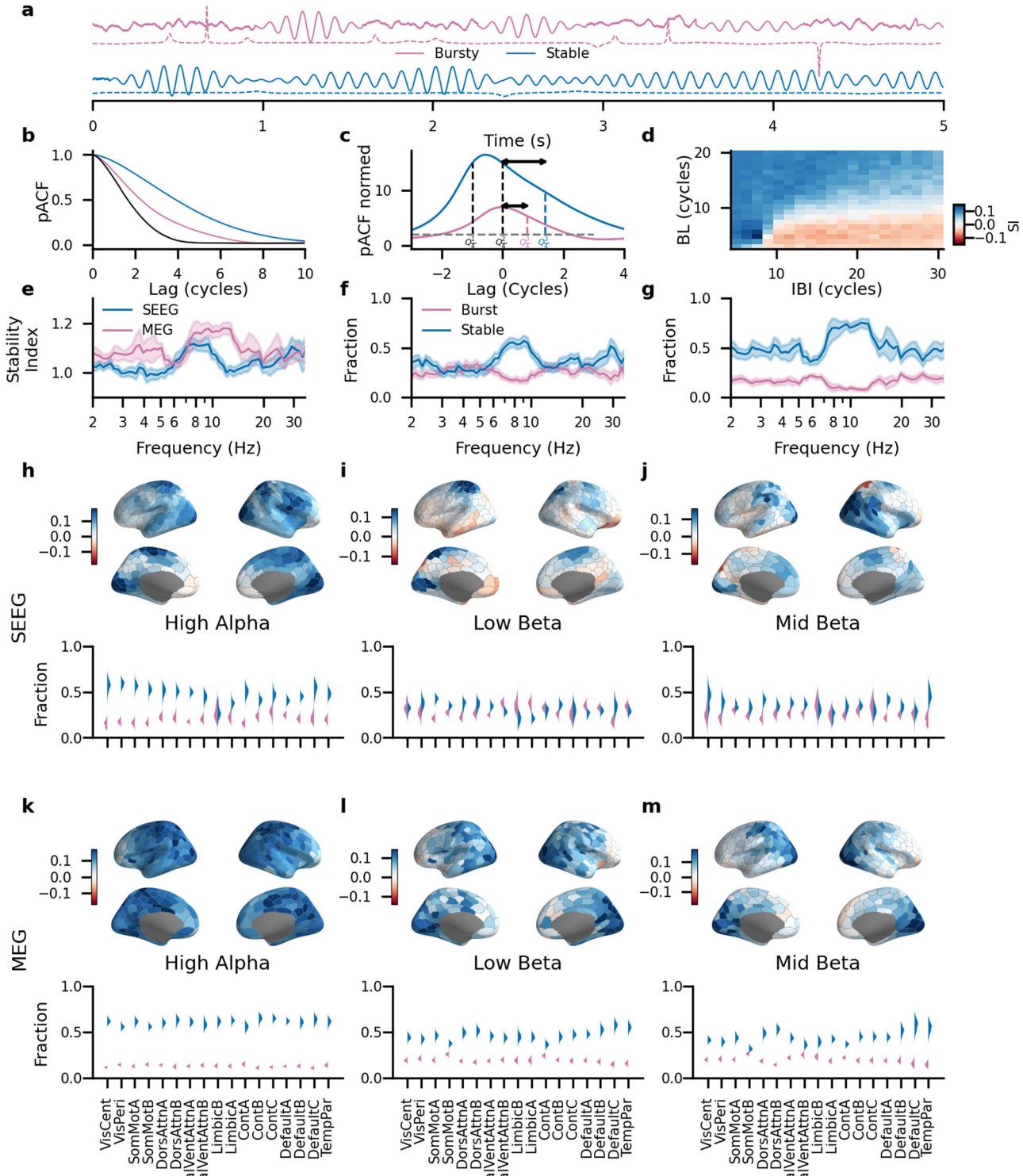

**Fig. 7 | Level of burst activity varies between frequencies and anatomical locations. a** Example of simulated burst-like (red) and stable (blue) activity, the dashed line indicates the instantaneous frequency of a signal. **b** pACF curves for the stable (blue), bursty (red), and noise (black) signals. **c** npACF curves for the bursty and stable signals. Despite having the same $Q\frac{1}{4}$ and $Q\frac{1}{2}$ the signals have different $Q\frac{3}{4}$. **d** Heatmap of stability index for the artificial data generated with varying burst length and interburst interval. **e** An average stability index across different frequencies for SEEG (blue) and MEG (red) data. The average value was estimated with bootstrapping (*N* rounds = 1000). Fraction of signals with stable (SI > 0.05) and burst (SI < −0.05) activity for SEEG (**f**) and MEG (**g**) data. Anatomy of the average stability index value and a fraction of stable and burst patterns aggregated across 17 systems for SEEG (**h**–**j**) and MEG (**j**–**m**) data for three frequency clusters detected previously: high alpha, low beta, and mid beta. Violin plots show distribution of the bootstrapped of each pattern (*N* rounds = 1000), the kernel size was computed using the Scotts method. Shaded areas indicate bootstrapped confidence intervals (5th and 95th percentiles, *N* bootstraps = 1000).

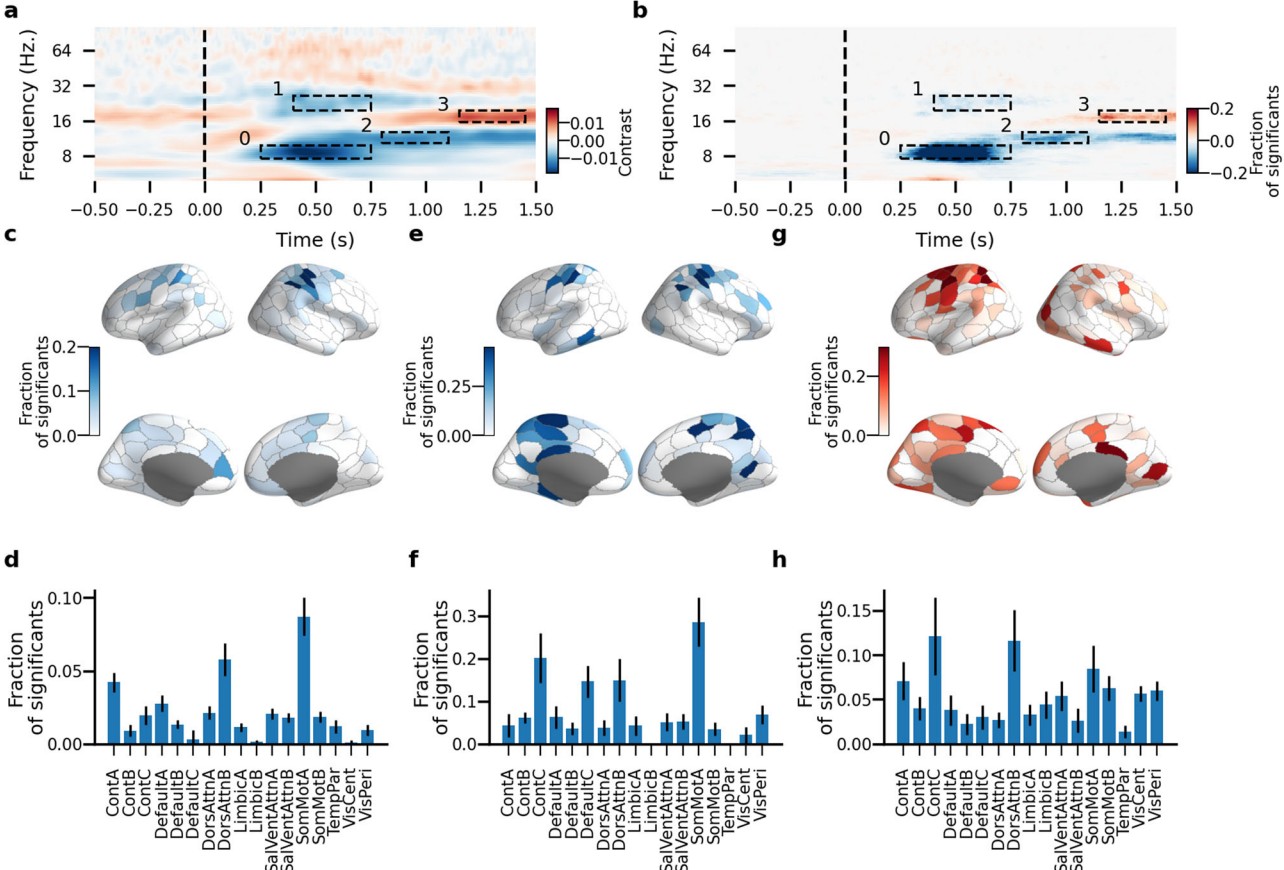

**Fig. 8 | Time-frequency phase autocorrelation representation during threshold-stimulus detection task.** Heatmap of the average pACF lifetime contrast values (**a**) and a fraction of significant differences (**b**) between two conditions (Hit–Miss) for each pair of frequency and parcel. Significance was estimated with surrogates using the shuffle test with random permutation of hit and miss labels (*N* shuffles = 10,000). Anatomy and bar plots averaged across functional systems of a fraction of significant values for the first (**c**, **f**), second (**d**, **g**), and third responses (**e**, **h**). Error bars in barplots represent a standard deviation of the mean value.

We establish here a new measure for characterizing and quantifying rhythmicity, the phase-autocorrelation function (pACF), that is based on phase stability and yields the "lifetime" of neuronal oscillations in an amplitude-independent manner. This approach is similar to the classical autocorrelation function that has been extensively used to quantify temporal stability in both broad- and narrow-band signals. The autocorrelation function, however, is based on real-valued signals and therefore the amplitude of each sample biases the weight of said sample in the autocorrelation function. For this reason, the ordinary autocorrelation function retains amplitude as a confounder that biases the estimates of oscillation lifetimes toward those of large-amplitude oscillations. This problem is greatly mitigated by pACF using only phase information and being independent of the amplitude (Supplementary Fig. 1h, i).

Using pACF, we found here that spontaneous human brain activity exhibits genuinely rhythmic oscillations only sparsely and with high anatomical and spectral specificity. We elucidated the cortical architecture of rhythmicity in both meso- and macro-scale neuronal oscillations, in SEEG and MEG data, respectively. We further established that pACF is a necessary requisite for long-range phase synchronization between cortical regions and that the approach can be extended to the detection of "burstiness" (i.e., how much of the observed signal is bursts). We also show that pACF could be adapted for the fine-grained time-frequency event analysis. These findings suggest that pACF is a powerful tool for the study of neuronal oscillations by quantifying rhythmicity in a direct and objective manner.

Indeed, explicit measurements of rhythmicity were commonplace in the early years of neural oscillations research[67,68], when both autocorrelation functions of continuous data[69,70] and histograms of spikes[71] were routinely used to quantify rhythmicity, while the power spectrum was used to detect the presence of oscillations. Over time, however, the use of classical autocorrelation-based methods has declined and power-spectrum approaches have become the primary way of analyzing oscillations in general, where oscillations are defined relationally as frequencies with greater power than neighboring frequencies.

In this approach, oscillatory power is qualified as a function of frequency and the concept of rhythmicity can only be indirectly operationalised as the width of the spectral peak, which can still be confounded by other signal properties. Here we advance a new approach to quantify rhythmicity explicitly and objectively using phase autocorrelations. While power spectral methods have also been extended to yield a rhythmicity-like measurement with "lagged coherence"[72], this approach embodies a frequency-shifting bias that leads to erroneous peak-frequency estimates, which in the present pACF is accounted for with instantaneous frequency correction (Supplementary Fig. 1a).

Using simulations and real MEG and SEEG recordings, we established here that rhythmicity operationalized with pACF lifetime and amplitude operationalized with signal power tap to different constructs and are theoretically and experimentally double-dissociable. Importantly, we found that also for neuronal oscillations in vivo, rhythmicity and amplitude were poorly correlated both in SEEG and MEG data (Fig. 5c, d), which suggests that rhythmicity and amplitude arise from partially distinct underlying neurophysiological and systems-level mechanisms. To further address the relationship between pACF and power spectral approaches and putatively distinct underlying mechanisms, we used partial correlations and found that PSD was more correlated with amplitude than rhythmicity (Fig. 5a, b). However, contrasting eyes-open and eyes-closed resting states, where alpha power increases with the closing of eyes ("Berger effect"), showed that this

effect was in fact driven more by a change in rhythmicity than in amplitude. These data thus constitute the first line of evidence that functional changes in the brain dynamics may be associated with the stability rather than amplitude of neuronal oscillations.

In comprehensive data-driven analyses, we observed significant rhythmicity in narrow-band oscillations throughout the neocortical surface so that on average, 66% of SEEG electrode contacts and 73% of MEG parcels in individual subjects exhibited significant oscillations as defined with pACF. The most dominant oscillations were in alpha and beta bands as found before with spectral analysis methods[2,30]. However, rhythmicity analyses showed that these dominant oscillations were in fact comprised of multiple sub-bands with unique anatomical profiles and very limited anatomical similarity across frequencies despite the widespread similarity in PSD data. The presence of narrow-band oscillations was further substantiated in single-subject data, suggesting that despite wide power spectral peaks, true oscillations are sparse and confined to narrow frequency bands.

Additionally, we found that pACF was more accurate in identifying oscillations than spectral-based methods in several respects. First, pACF exhibited better frequency resolution (Fig. 2e, f) than PSD and dissected each of the alpha and beta bands into three sub-bands (Fig. 3c), while with PSD a single band was observed (Supplementary Fig. 3b). Second, pACF revealed sparsely anatomically located oscillations (Fig. 3d, e and Supplementary Fig. 3e, f). Thus, in addition to being a qualitatively novel observable, pACF also yields insight into the architecture of oscillations by overcoming the intrinsic resolution limitations of the power-based analysis. This might be because a purely single-frequency signal appears as a peak in FFT only when it has constant amplitude and phase, while fluctuations in amplitude and discontinuities in phase lead to non-zero FFT coefficients in neighboring frequency bins[73]. This phenomenon is known as "frequency leakage" and leads to oscillatory processes having spectral peaks with a width that is dependent on their amplitude[39] and phase fluctuations.

Neural oscillations exist at multiple time scales and canonically are divided into frequency bands that reflect specific cognitive or physiological processes[74] such as theta (4–8 Hz), alpha (8–15 Hz), or beta (15–30 Hz) bands. Our single-subject data suggested that despite wide power spectral peaks, true oscillations could be sparse and confined to narrow frequency bands. At the group level, we indeed found rhythmicity to splice the canonical frequency bands into multiple sub-bands with unique anatomical profiles and very limited anatomical similarity across frequencies (Supplementary Fig. 3c) despite the widespread similarity in PSD data (Supplementary Fig. 4c). This constitutes the first line of evidence that rhythmicity-based characterization yields novel insight into oscillatory brain architecture.

The most stable oscillations were located in attentional and sensory systems, while in the limbic system and anterior frontotemporal cortex (Fig. 4b), oscillations were the rarest and most short-lived. In line with prior PSD-based results[35] and with the classical definition of alpha and mu rhythms[45], alpha-band oscillations were strongest in posterior brain regions, while low beta-band oscillations were located centrally around sensorimotor regions. The present findings, however, also revealed variability among the topographies of nearby frequencies and revealed a nested structure of alpha and beta bands with low-, mid-, and high-frequency communities (Fig. 3d, e).

The gradient-like organization of many aspects of brain architecture has recently emerged as a central topic in neuroscience[75]. It has been proposed that an underlying macroscopic gradient-like organization shapes brain dynamics[76–78]. Power-spectral-peak frequencies of neuronal oscillations exhibit both a medial-to-lateral and a posterior-to-anterior hierarchy gradient of increasing alpha-peak frequency[30,79]. Here, we revealed the anatomy of peak frequencies of genuine oscillations which was complex and distinct for individual frequency bands. Importantly, unlike the more gradient-like prevalence maps, the maps of peak frequencies lacked salient directions in principal axes (for an alpha, Fig. 4h, i, n, o; beta, see Supplementary Fig. 4c, d, g, h).

Meanwhile, activity in the prefrontal cortex (PFC) is dependent on neurochemical control through noradrenaline and dopamine, or of acetylcholine[80], which influence oscillation dynamics as shown by pharmacological modulation[81] and genetic variability[82]. Moreover, the anterior PFC, which is thought to be a largely homo-sapiens specific and evolutionarily the area with most recent expansion in humans that plays a role in higher cognitive functioning, was largely devoid of oscillations in our data. It is possible that PFC shows intrinsic oscillatoriness only in the higher cognitive processing functions, such as during working memory processing, while it is silent in resting state[51].

As the pACF approach yielded narrow oscillatory peaks, it allowed us to identify oscillations in individual bands in a more precise way. We quantified the fraction of cortical areas that exhibited oscillations in only one or two frequencies. Strikingly, in both SEEG and MEG, less than 30% of areas with oscillations exhibited them in two or more bands, and <10% in more than two. The neocortical surface is thus largely "mono-oscillatory" with the dominant resting-state oscillations almost exclusively in the theta/alpha band (Fig. 4g, m). Conversely, significant beta oscillations were found to largely only co-occur with theta/alpha oscillations. Moreover, the anatomical profiles of single- and dual-frequency band oscillations were distinct (Fig. 4d–f) and the beta-coupled theta/alpha oscillations exhibited systematically higher frequencies than those found alone (Fig. 4g–i).

Inter-areal synchronization of neuronal oscillations is thought to regulate communication across the cortex. However, there has been scarce prior research into its relationship with the oscillatory properties of individual regions. At the meso-scale, our results suggest that oscillations at the same frequency are required for significant phase synchronization. We found positive correlations between average node synchrony and oscillation lifetime (Fig. 6b, f, g) and observed that significant phase synchrony can be predicted from the rhythmicity of the coupled areas (Fig. 6d, h, l) in the model as well as in SEEG and MEG recordings.

Communication between neurons is based on neuronal spiking which occurs in different patterns from rhythmic spiking to short sequences of fast activity called bursts[83]. Single-cell activity is not prominent enough to be recorded with extracellular potentials such as SEEG, but the activity of multiple densely connected cells might cause detectable signals[84]. Recent primate electrophysiology research has shown that working memory demands might be dependent on burst-like activity rather than stable oscillations in the prefrontal cortex[51,85]. It also has been proposed that brain dynamics in general are composed mostly of transient bursts rather than stable oscillations[48]. Such bursts are typically defined as short periods of high-amplitude narrow-band activity[86].

Methodological approaches to quantifying bursting activity and measuring burstiness have, however, remained an area of active development. Power-spectral methods cannot dissect stable and burst-like activity with a poor signal-to-noise ratio, hence the solutions have remained predominantly based on amplitude time-series analyses. These approaches match the visual intuition of bursts often exceeding the "background activity" in amplitude, but leave implicit the notion of bursts possibly also pertaining to the temporal stability of neuronal oscillations.

In this work, we utilized pACF and defined bursts as short periods of highly stable oscillations separated by activity without long-range correlations (Fig. 7a). This is reflected as high autocorrelation at the start of the pACF curve but a sharp decline in the middle, resulting in the shape of the pACF curve skewing to the right (Fig. 7b). Using this approach, we showed that while overall, alpha-band oscillations were the most stable, beta oscillations showed the highest fraction of recordings with high "burstiness" (Fig. 7e–g). The findings of burstiness specifically in beta-band are intriguing and in line with previous results in humans[87] and from the monkey prefrontal cortex during working memory[51].

Finally, to address whether rhythmicity can be assessed in event-related research paradigms, we analyzed MEG data from a visual threshold-stimulus-detection task where identical, very weak stimuli are detected half of the time[9]. Cluster analysis of pACF lifetimes showed that rhythmicity is able to detect the previously established difference between hit and miss

TSDT responses in theta (4–6 Hz), alpha (8–10 Hz), and beta (20–27 Hz) with higher resolution than the classical methods. In addition, it also revealed a highly stable period in the high-alpha frequency band (14–16 Hz) which was not detected with classical methods to analyze event-related data. Furthermore, pACF revealed contrasts at a higher frequency resolution and evoked responses in more frequency bands than the induced response approach. pACF can thus reveal insight also into event-related dynamics.

To summarize, pACF analyses of meso- and macroscale neuronal oscillations show that rhythmicity is complementary to signal power, yields unique information, and allows for precise investigation of brain oscillatory architecture. It also may be used to estimate the burstiness of a signal and for fine-grained time-frequency analysis of visual stimulus processing. These findings indicate that pACF may be a valuable tool in a wide range of neuroscience scales and studies.

## Methods
### SEEG data acquisition
Using stereoelectroencephalography (SEEG), we acquired an average of 10 min of uninterrupted spontaneous activity with eyes closed from 92 consecutive patients affected by drug-resistant focal epilepsy undergoing presurgical evaluation. From this cohort, 25 patients were excluded due to previous brain surgery or large cortical malformations identified from magnetic resonance images (MRI). Additionally, 6 subjects were excluded because of recording artefacts. After the exclusion of electrode contacts in the epileptic zone (EZ), the final cohort of 61 patients (age: 29.9 ± 9.7, 35 male) yielded a total of 6945 gray matter contacts (113 ± 16 per subject, mean ± standard deviation, one session per subject).

We acquired monopolar (with shared reference in the white matter far from the putative epileptic zone) local-field potentials from brain tissue with platinum-iridium, multi-lead electrodes. Each multi-lead electrode consisted of up to 15 contacts, each measuring 2 mm in length, 0.8 mm in thickness, and spaced 1.5 mm apart from border to border (DIXI medical, Besancon, France). The anatomical positions and amounts of electrodes varied exclusively according to surgical requirements[88]. On average, each subject had 17 ± 3 (mean ± standard deviation) shafts (range 9–23) with a total of 153 ± 20 electrode contacts (range 122–184, left hemisphere: 66 ± 54, right hemisphere: 47 ± 55 contacts, gray-matter contacts: 113 ± 16.2).

Prior to electrode implantation, the participants provided written informed consent to participate in research studies and for the publication of their data. This study was approved by the ethical committee (ID 939) of the Niguarda Ca' Granda Hospital, Milan, and was performed according to the Declaration of Helsinki.

### MEG data acquisition—resting state
Ten minutes of eyes-open resting-state data were recorded with a Triux 306-channel MEG (Elekta-Neuromag/MEGIN, Helsinki, Finland; 204 planar gradiometers and 102 magnetometers) at BioMag Laboratory, HUS Medical Imaging Center, from 54 healthy participants (26 females; mean age 31.27 ± 9.16, median age 28.5, the average number of recordings per subject 3.9, min 1, max 18, the analysis outcome was average across sessions of each subject) and 15 healthy participants for the eyes-closed and eyes-closed cohorts, no participants were excluded. Participants were seated in a dimly lit room and instructed to focus on a cross displayed on the screen in front of them. Bipolar horizontal and vertical EOG were recorded for the detection of ocular artifacts. MEG and EOG were recorded at a 1000 Hz sampling rate.

T1-weighted anatomical MRI scans (MP-RAGE) were obtained for head models and cortical surface reconstruction at a resolution of 1 × 1 × 1 mm with a 1.5 Tesla MRI scanner (Siemens, Munich, Germany) at Helsinki University Central Hospital. The study protocol for MEG and MRI data was approved by the Coordinating Ethical Committee of Helsinki University Central Hospital (ID 290/13/03/2013), written informed consent was obtained from each participant prior to the experiment, and all research was carried out according to the Declaration of Helsinki.

### MEG data acquisition—TSDT
Participants performed a visual Threshold-Stimulus Detection Task (TSDT) while neural activity was recorded with a Triux 306-channel MEG (Elekta-Neuromag/MEGIN, Helsinki, Finland; 204 planar gradiometers and 102 magnetometers) at BioMag Laboratory, HUS Medical Imaging Center, from 23 healthy participants (15 females, mean age: 33, range: 18–57, 1 left-handed, one session per subject). All participants had to meet the following inclusion criteria: be between 18 and 60 years old, have normal or corrected-to-normal vision, and be compatible with MEG and MRI. Exclusion criteria included any neurological or neuropsychiatric disorder. Bipolar horizontal and vertical EOG were recorded for the detection of ocular artifacts. MEG and EOG were recorded at a 1000-Hz sampling rate. A total of 18 subjects were used for the analysis.

During the TSDT task, participants were seated in a dimly lit room and presented with slowly varying dynamic Perlin noise, which covered a visual angle of 10 degrees and was projected onto a screen inside the magnetically shielded room. After a variable inter-stimulus interval (1.5–4.5 s), one of two geometrical shapes was presented at the center of the Perlin noise (1° visual angle). To find the individual visual detection threshold, a QUEST staircase adaptation procedure was performed to achieve a 50% detection rate. Subjects were instructed to give an answer with a response device if they detected the stimulus. The answer hand was randomized for each subject. A total of 500 trials were collected during a 25-min MEG measurement.

The study protocol for MEG and MRI data was approved by the Coordinating Ethical Committee of Helsinki University Central Hospital (ethical number 290/13/03/00/2013), written informed consent was obtained from each participant prior to the experiment, and all research was carried out according to the Declaration of Helsinki.

### SEEG data preprocessing
The closest white-matter (cWM) referencing scheme was used for SEEG data. Neocortical electrodes in gray matter were referenced to their nearest contacts in white matter[23]. This referencing scheme produces signals with consistent polarity and limits the mixing of signals from active sources, resulting in more accurate phase estimates. Prior to the main analysis, cW-referenced SEEG time series were low-pass filtered with FIR filter with a cutoff at 440 Hz and stop-band at 500 Hz (60 Hz transition band, −6 dB suppression at 475 dB).

### MEG data preprocessing
We used Maxfilter with temporal signal space separation (tSSS) (Elekta Neuromag Ltd., Finland) to suppress extra-cranial noise in sensors and to interpolate bad channels. The data were low-pass filtered at 249 Hz and Notch-filtered to remove line noise and the corresponding harmonics. We then used independent component analysis (MNE, https://mne.tools/stable/index.html) and visual inspection to identify and remove components that were correlated with ocular (using the EOG signal), heart-beat (using the magnetometer signal as reference), or muscle artefacts.

We used the FreeSurfer software (surfer.nmr.mgh.harvard.edu) for volumetric segmentation of MRI data, surface reconstruction, flattening, cortical parcellation, and neuroanatomical labeling with the 400-parcel Schaefer atlas[44] that favors functional network topology over structural (gyral) topology, with each cortical parcel assigned to one of the functional systems defined in ref. 89.

We used the MNE software package[90,91] to create head conductivity models and cortically constrained source models with 8000–14,000 sources per hemisphere (spacing of 5 mm) and subject, for MEG-MRI co-localization, and for the preparation of noise covariance matrices (from frequency band 151–249 Hz, 800–500 ms before stimulus for each trial), forward and inverse operators. We created fidelity-weighted inverse operators for optimized reconstruction accuracy[28,92] and collapsed source time series to the 200 parcels of the Schaefer atlas.

## Signal preprocessing

In this study, we used complex Morlet wavelets to obtain a narrow-band representation of a signal[93]. Line-noise harmonics were suppressed with a notch filter centered at 50 Hz with 53 dB suppression and 1 Hz band transition widths. The low-pass filtered data were then separated into narrow frequency bands with 81 equally log-spaced ($f_{i+1} = f_i*1.05$) Morlet wavelets with frequencies ranging from 2 Hz to 100 Hz. The upper frequency boundary was selected to keep the minimum number of samples per cycle not less than 10 and the minimum pACF step of 0.1 cycles. We used Morlet Wavelet with 7.5 cycles to achieve a detailed frequency resolution but keep time resolution as good as possible (Supplementary Fig. 1b–d).

## DWI and structural connectome

We computed structural connectomes for 57 unrelated subjects randomly selected from the WU-Minn 1200 subjects dataset of the Human Connectome Project (HCP; https://www.humanconnectome.org/). For each subject, White Matter (WM) tracts were reconstructed from preprocessed Diffusion Weighted Imaging (DWI) data with MRtrix3 (https://www.mrtrix.org/). In summary, from each subject's pre-processed 3T DWI data, we estimated the Multi-Shell Multi-Tissue (MSMT) response and performed MSMT spherical deconvolution[94] and probabilistic tractography to generate a preliminary tractogram of 50 million streamlines (maximum tract length = 250 mm; Fiber Orientation Distribution (FOD) amplitude cutoff, 0.01; seeding from the gray matter-white matter interface; application of the Anatomically Constrained Tractography (ACT) framework[95]).

We then filtered this initial tractogram with the Spherical-deconvolution Informed Filtering of Tractograms (SIFT) framework[96] in order to produce a more biologically plausible version of the tractograms and to reduce the bias for longer, thicker tracts inherent to the tracking algorithm[97]. The resulting final tractograms for each subject were composed of 5 million streamlines. For each subject, we created an individual cortical parcellation (400 parcels) based on the Schaefer atlas[44] and then collapsed streamlines to weighted edges between parcels. Each endpoint of a streamline was matched with the most likely parcel using a radial search spanning 3 mm outward. The weight of the connection between any two parcels was determined by the count of streamlines linking them. Self-connections, represented on the diagonal of the matrix, were assigned a weight of zero.

## Phase autocorrelation estimates

To estimate the phase autocorrelation function of an analytical signal (e.g., after wavelet-filtering), we calculated the Phase Locking Value (PLV) of a signal with a delayed copy of itself. The PLV is the absolute value of the complex PLV (cPLV)[31]. The population pACF was thus defined as:

$$\text{pACF}(l) = E[\text{CS}_{x,x(l)}]$$

where $E[.]$ denotes the expected value and $\text{CS}_{x,x(l)}$ denotes the cross-spectrum between a complex signal $X$ and a copy delayed with lag $l$. In practice, pACF is estimated using limited data and sample pACF is defined as:

$$\overline{\text{pACF}(l)}_{\text{sample}} = \frac{1}{N} \sum_{i=1}^{N} \text{CS}_{x,x(l)}$$

where $N$ is the total number of samples.

pACF was computed for a range of lags of the corresponding signal ranging from zero lag to 20 cycles with a step of 0.1 cycles. Note that pACF will be a delta function if signal phases are independent and a constant if signal phases are linearly dependent. However, narrow-band filtering induces artificial phase-autocorrelations into a data that are dictated by the time domain width of the filter. In the case of Morlet wavelets, the strength of these autocorrelations depend on the number of cycles in the wavelet (Supplementary Fig. 1c).

In the common approach, the transformation of lag in cycles to timesamples would be done by multiplying it with the sampling frequency and dividing on the frequency of interest $lag * \frac{\text{SRate}}{\text{Frequency}}$. However, this approach biases the pACF spectral peak toward higher frequencies than the frequency of the ground truth oscillation as well as the PSD peak (Supplementary Fig. 1a).

In order to overcome this issue, we applied the instantaneous frequency correction where we divided the sampling frequency on the sample mean instantaneous frequency of the wavelet-filtered data rather then its central frequency $lag * \frac{\text{SRate}}{\overline{\text{IF}}}$ where $\overline{\text{IF}}$ is a mean instantaneous frequency.

To take into account possible long-lasting autocorrelations with low similarity, we transformed the pACF function to an explained variance function by dividing by the sum over all lags. The lifetime of the pACF function is defined as the first point where cumulative pACF is higher than a given value (we used a threshold of 0.9 in this study Fig. 1d).

An interesting feature of pACF, stemming from the fact that it uses only phase information, is that it does not assume that a signal is stationary. Indeed, analyses of simulated data corroborate this and show that the pACF spectra are the same for stationary and non-stationary signals (Supplementary Fig. 1g).

## Functional connectivity estimation

To estimate inter-areal phase interactions at individual subject levels, we computed synchrony metrics for each pair of contacts (SEEG) or parcels (MEG). For SEEG, we used the phase locking value (PLV, see "Phase autocorrelation" section), and for MEG data, the weighted Phase Lag Index (wPLI) which is defined as:

$$\text{wPLI} = \frac{| \sum_{i=0}^{N} \text{imag}(\text{CS}_{x,y})|}{\sum_{i=0}^{N} |\text{imag}(\text{CS}_{x,y})|}$$

where $\text{imag}(\text{CS}_{x,y})$ is the imaginary part of the cross-spectrum of the complex time series $x$ and $y$[98,99].

## Data modeling

We validated the pACF approach on synthetic data with known oscillatory properties. To simulate this data, we computed the sum of pink noise and an oscillatory component. To generate an oscillatory component with varying longevity, we filtered the pink noise signal several times (from 1 to 10) first with a frequency-tolerant Morlet wavelet (number of cycles = 3.5) and then with a time-tolerant Morlet wavelet (number of cycles = 7.5). After each round of filtering, we normalized a component by dividing it by its mean envelope to preserve the amplitude magnitude. To obtain a signal with varying power, we computed the sum of pink noise and the real part of the same pink noise signal filtered with a Morlet Wavelet and multiplied it with a coefficient (from 5 to 15).

To evaluate the relationship between phase autocorrelation and phase synchrony on simulated data, we used a Kuramoto Model[100]. The model was adapted to have several "nodes", each of them a population of 500 oscillators. The phase dynamics for each oscillator are given by:

$$\frac{\delta \theta_i^m}{\delta t} = \omega_i^m + \frac{K_{\text{intl}}}{N} \sum_{j=0}^{N} \sin(\theta_i^m - \theta_j^m)$$
$$+ \sum_{j=0}^{M} W_{j,m} * \sin(\theta_i^m - \Theta_j) + \mathcal{N}(0, \sigma^2)$$

where $\theta_i^m$ is the phase of $i$th oscillator in $m$th node, $\omega_i^m$ is the natural frequency of the oscillator, $K_{\text{int}}$ is the coupling coefficient within a node, $N$ is the number of oscillators within a node, $M$ is the total number of nodes, $W_{i,j}$ is the connection strength between nodes $i$ and $j$ and $\Theta_j$ is the average phase of the $j$th node. Strengths of inter-node connections were derived from structural human connectome based on Schaefer parcellation[44] with the

resolution of 400 parcels. The raw values of the connectome were log-transformed, and $K$ was set to 10.0 for all nodes.

### Spectral clustering of pACF profiles

To group pACF and PSD spectra by their similarity, we used a hierarchical algorithm based on Ward's method[101]. This algorithm initially assigns to each sample a unique cluster and sequentially merges two sub-clusters with the minimum distance to a new one. Following the method, the distance between two clusters is defined as:

$$\Delta(A, B) = \frac{n_A * n_B}{n_A + n_B} * || \overrightarrow{m_A} - \overrightarrow{m_B} ||^2$$

where $n_A$ and $n_B$ are number of samples in clusters A and B, $\overrightarrow{m_A}$ and $\overrightarrow{m_B}$ are centroids of these clusters.

To detect frequency bands of interest, we first computed the Pearson correlation coefficient between vectors of pACF parcel values for each pair of frequencies. We then transformed the correlation coefficient to the similarity index $s$ using the next formula $s = \frac{\sqrt{(1-\rho)}}{\sqrt{(2)}}$ where $\rho$ is the Pearson correlation coefficient (Supplementary Fig. 3b). Then we constructed a weighted similarity graph where each edge's weight indicates the spatial similarity between two frequencies and removed all edges with non-significant distance (obtained with one-sided permutation test, $N = 1000$, 99th percentile of the null-distribution was considered as a significance threshold). To detect overlapping communities in such a graph, we applied a Deep Non-negative Matrix Factorization method[102] and selected communities with an average fraction of significant activity of more than 5%.

### Oscillatory peak detection and analysis

To detect peaks of oscillatory activity, we first linearly interpolated a spectrum and applied the peak detection algorithm from the Scipy library[103] to find peaks with a minimum height of the noise level, minimum width of 1 Hz, and distance to the closest next peak of at least 3 Hz.

We pooled detected peak frequencies to parcels and computed the fraction of electrode contacts without oscillatory components at all, with one, two, or more peaks. To find the central frequency of a parcel, we pooled central frequencies of individual signals that belong to the same parcel to a histogram, divided it into alpha (8–15 Hz) and beta (15–30 Hz) frequency bands, and applied the peak detection algorithm to the frequency-related parts of the histogram. If several peaks were detected in a band, we considered their median as a central frequency.

### Stability index

To compute the stability index we at first compute $\text{npACF} = \frac{\text{pACF}}{\text{pACF}_{noise}}$—a pACF normalized by computing ratio with corresponding noise-level pACF curve (Fig. 7c). As the next step, we extract the longest segment of npACF with values higher than a given threshold (in this analysis we used the threshold of 2) and detected $Q_1$, $Q_2$ and $Q_3$ values where $Q_k$ value represents the $k$th quantile. As the last step, we compute the skewness of the segment and define the stability index as:

$$\text{SI} = \frac{Q_3 + Q_1 - 2 * Q_2}{Q_3 - Q_1}$$

### Time-frequency representation of the phase autocorrelation function

The estimation of pACF is based on the phase difference between individual samples which can be used to construct the time-frequency representation of a signal. In order to do so, we computed PLV in moving windows of size 2.5 cycles between a signal and a delayed version of itself for lags of 1 to 3 cycles with increments of 0.1 cycles and averaged PLV across lags. The time-resolved pACF was estimated as:

$$x_t = \sum_{l=0}^{L_{max}} \frac{\sum_{i=t-W/2}^{t+W/2} z_i^l}{W} / N$$

where $t$ is a timestep, $l$ is a lag, $W$ is a window size in samples, $N$ is the number of lags, and $z$ is the phase difference between a signal and its delayed version.

### Statistics and reproducibility

To test whether a pACF lifetime was significantly different from that of a random process it is essential to take into account the filter-induced auto-correlations and to match the real-data scaling exponents (Supplementary Fig. 1e, f). We wavelet-filtered 10,000 realizations of the pink-noise data using the same wavelet parameters and matching the data length, applied the pACF lifetime pipeline and computed the 99th percentiles of the pACF lifetime values for each frequency.

In order to test whether pACF lifetime is significantly correlated with average node synchrony and whether pACF predicts presence of significant synchrony between channels, we randomly shuffled pACF values ($N = 10,000$) and computed the distribution of correlation coefficients for the surrogate data. We used 5th and 95th percentiles as significance thresholds for the correlation (negative and positive) and 95th percentile for the predictivity.

We used a similar permutation approach to test the significance of TSDT response values. We estimated the null hypothesis distributions with surrogates using a two-sided permutation test. We randomly permuted hit and miss labels ($N$ shuffled = 10,000), computed the difference between response variables (pACF lifetime, PLF and amplitude response) and used the 5th and 95th percentiles of the distributions as the significance threshold. We used the F-max statistic to correct for multiple comparisons.

### Reporting summary

Further information on research design is available in the Nature Portfolio Reporting Summary linked to this article.

### Data availability

Raw data cannot be made available due to data privacy regulations set by the ethical committees. The data underlying results to reproduce the main findings is deposited in the DataDryad repository[104]. Other types of data can be shared upon reasonable request.

### Code availability

For obtaining the main findings and computing phase autocorrelation function of any data can be found on Github (https://github.com/palvalab/discovering_rhythmicity).

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

## Acknowledgements
We thank Hamed Haque, Jonni Hirvonen, Sami Karadeniz, Salla Markkinen, Santeri Rouhinen, and Jaana Simola for help with the MEG recordings and Annalisa Rubino for the help with the SEEG recordings. This study was supported by the Academy of Finland (J.M.P., project numbers: 253130, 256472, 281414, 296304, 266745; S.P., 266402, 266745, 303933, 325404), by the Juselius Foundation (J.M.P., S.P.), by the Helsinki University Research Funds (J.M.P., S.P.), and by the Finnish Cultural Foundation (G.A., 12938).

## Author contributions
V.M. and J.M.P. conceived the study; V.M. wrote the analysis code; V.M. conducted the analysis; G.A. provided clinical information and preprocessed SEEG data; F.S. provided preprocessed resting-state MEG data; J.J. provided preprocessed TSDT data; V.M., J.M.P., G.A., F.S., J.J., and S.P. interpreted the data and contributed to the writing and revising of the manuscript. All authors read and approved the manuscript.

## Competing interests
The authors declare no competing interests.
