## [Peer review file · Communications Biology]

Reviewers' comments:

Reviewer #1 (Remarks to the Author):

Thank you for the opportunity to review "Rhythmicity of neuronal oscillations delineates their cortical and spectral architecture" by Myrov and colleagues. The manuscript describes a simple and intuitive method to quantify the presence of oscillations using phase autocorrelation. Using both simulations and empirical data, the authors demonstrate that the method captures a meaningful feature of neural activity. The manuscript is extensive, showing how the method can be applied to multiple questions and compared to more traditional PSD findings. The work is also rigorous and I applaud the authors for making the code available. I just have two minor suggestions to improve what is already an excellent manuscript:

- The anatomical findings are fascinating, particularly how rhythmicity seems to be so spatially circumscribed. Could the authors speculate why some areas appear devoid of oscillations, e.g. frontal cortex? Is it because of some specific projection patterns, or because of some micro-architectural features?

- Page 20. The work does not actually use "DTI", I'd advise the authors to change the subsection title to "DWI"

Reviewer #2 (Remarks to the Author):

In the submitted manuscript, a new index for quantifying the level of rhythmicity of electrophysiological brain signals is introduced and applied to MEG and intracranial EEG recordings of human participants. This index is derived from the autocorrelation function of the instantaneous phases of the signal and is referred to as the phase autocorrelation function (pACF) in the manuscript. The index itself is referred to as the lifetime of the phase autocorrelations. Although this is an interesting approach, there are several methodological issues, unaddressed control analyses, and the claims are not completely corroborated by the analyses results (details below).

The manuscript is well-structured but poorly written: At many places in the text, it is difficult to find out exactly what was done. Furthermore, several phrases and statements are unclear, unnuanced, or simply incorrect. Some examples are given in the minor comments below, but there are several more that I didn't list.

The method is not well worked out theoretically and there seems to be a fundamental problem with hypothesis testing for significant rhythmicity based on the proposed index (see major comment 2). Specifically, the null hypothesis is left implicit and when made explicit, it becomes clear that it does not coincide with the desired null hypothesis (the absence of rhythmicity) but with the absence of (ordinary) autocorrelations.

MAJOR COMMENTS

1. The advantage for quantifying the level of rhythmicity of a signal in terms of the pAFC instead of the power spectrum is poorly explained. In the Introduction (Line 53) the authors write

“However, being amplitude-based, this approach is only a qualitative indicator of rhythmicity and does not quantify it explicitly”.

First, being amplitude-based and being qualitative are independent issues. Amplitude-based measures of rhythmicity can be both qualitative and quantitative and this holds for any other measure, whether based on amplitudes or not. Second, it is unclear what the authors mean with the phrase “quantify it explicitly”. What does it mean to quantify something implicitly? The authors go on to write:

“For example, brief high-amplitude bursts have been shown to be a robust neuronal phenomenon and to play a role in cognitive functions but can not be dissociated from sustained and temporally sustained oscillations with power-spectral amplitude-based methods.”

The claim here is that amplitude-based methods cannot distinguish between transients and sustained oscillatory dynamics, but the exact same thing can be said about the pACF. Both methods are based on averaging over windows during which the signal is assumed to be stationary. Furthermore, the issue of non-stationarity is entirely unrelated to whether a measure is qualitative or quantitative and hence it is unclear why it is used as an example in this context.

I do understand that if one is interested in phase-dynamics of the signal, it is more natural to consider the pACF than the power spectrum, because the pACF is defined solely in terms of phases and doesn't refer to amplitudes (but see major comment 2).

2. In the manuscript, “rhythmicity” is defined implicitly by the way the null-data is constructed. When it is made explicit, it becomes clear that “rhythmicity” is the same as “having (ordinary) autocorrelations”.

The null-data used by the authors is white noise, which is characterized by its autocorrelation function being a delta function at lag zero and this is equivalent to its power spectrum being flat. Hence the null-hypothesis is that, in the considered frequency range, the power spectrum is flat. This shows that the null-hypothesis cannot be expressed solely in terms of phases and that amplitudes need to be taken into account. More importantly, if the null-hypothesis is rejected, one can draw the conclusion that the power spectrum is not flat, i.e., the observed signal is not white-noise. This conclusion is very different from the desired conclusion which is the presence of rhythmicity.

3. The authors mention that most studies on electrophysiological signals use spectral instead of phase-based measures to characterize rhythmicity. The reason for this, I think, is that electrophysiological signals are nearly Gaussian and all information about a Gaussian signal is contained in its power spectrum. In particular, the pACF can be expressed in terms of the power spectrum and hence does not add information, unless the signal is non-Gaussian. I think this point should be made explicit early in the manuscript. To demonstrate that the pACF actually adds information, one has to show that some aspect of the pACF (in this case the lifetime) is unlikely to be drawn from the distribution of lifetimes under the Gaussian null-hypothesis (as quantified by a p-value). The null-distribution of lifetimes can be approximated by calculating the lifetimes of a large number of phase-randomized copies of the observed signal.

MINOR COMMENTS

1. At line 301 the authors write

“Neuronal oscillations encompass a wide range of neuronal processes where the phase of the oscillation is the key mechanistically and functionally significant element.”

This is a significant statement that should at least be backed up by references. It is also true that amplitude fluctuations play an important role in brain function, given that the resting-state networks reflect correlations between amplitudes and not phases. I therefore think that the above statement should be nuanced.

2. It will be good if the authors distinguish between the population and sample pACF, at least in the methods section. Also, when defining the pACF in the methods section, the authors might want to add that the lifetime is zero if and only if the instantaneous phases at different time-points are independent (the pACF will then be a delta function).

3. At time 321 the authors write

“Explicit measurements of rhythmicity used to be, in fact, commonplace in early years of research on neuronal oscillations when both autocorrelation functions of continuous data and histograms of spiking were routinely used to quantify rhythmicity. Recently, however, the methodological focus has turned to power spectral approaches...”

First, power spectral methods are not recent but are, in fact, the classical way to analyze (neuronal) signals. Second, being Fourier transforms of each other, the autocorrelation function and the power spectrum provide the same information. It is therefore puzzling to me why the authors contrast the two.

4. At line 325 the authors write

“We advance here a new approach to quantify rhythmicity explicitly and objectively with phase autocorrelations.”

This gives the impression that spectral methods are, somehow, not objective. Please phrase this more carefully and accurately.

5. The authors define the “lifetime” of the phase autocorrelations and note that this is “similar to how exponential decay is indexed by a decay time constant”. However, for the exponential decay, there is a natural time-constant, whereas this is not the case for the pACF. It will be good to mention this.

6. The distribution of the null-data is not mentioned. In any case, it will be different from that of the observed signals, which is undesirable, because rejection of the null-hypothesis can then be due to the observed signal having a different distribution than the null-data and not due to the presence of rhythmicity.

7. On line 327 the authors mention that quantifying rhythmicity as done in ref. [56] has a “frequency-shifting bias that leads to erroneous peak-frequency estimates”

and then refer to Supplementary Figure 1a. However, this figure is difficult to understand and so it remains unclear what the above actually means. For instance, the legend of panel a reads “Morlet wavelets ...” but no wavelets are shown and it is unclear how it relates to wavelets. Please explain the above bias-issue and make the supplementary information self-contained.

8. On line 330 the authors write

“Using simulations, we established here that rhythmicity and amplitude, as operationalized with pACF lifetime and signal power, are theoretically and analytically double-dissociable.”

I’m not sure what this sentence means. Are the authors referring to the fact that the pACF is independent of the power of the signal? Furthermore, what does “analytically” mean here and how is it different from “theoretically”?

Point-by-point replies to Editorial and Reviewer comments on
“Rhythmicity of neuronal oscillations delineates their cortical and spectral
architecture”

By Myrov *et al.*

We thank the Editor and Reviewers for prompt handling of our manuscript and, in particular, for excellent and thorough comments. Please find below our point-by-point replies (**R**) and actions (**A**). We have done our best to rigorously and comprehensively improve the manuscript following the Reviewers’ suggestions.

Reviewer #1 (Remarks to the Author):

Thank you for the opportunity to review “Rhythmicity of neuronal oscillations delineates their cortical and spectral architecture” by Myrov and colleagues. The manuscript describes a simple and intuitive method to quantify the presence of oscillations using phase autocorrelation. Using both simulations and empirical data, the authors demonstrate that the method captures a meaningful feature of neural activity. The manuscript is extensive, showing how the method can be applied to multiple questions and compared to more traditional PSD findings. The work is also rigorous and I applaud the authors for making the code available. I just have two minor suggestions to improve what is already an excellent manuscript:

(Reply) We thank Reviewer #1 for their constructive and insightful comments!

- The anatomical findings are fascinating, particularly how rhythmicity seems to be so spatially circumscribed. Could the authors speculate why some areas appear devoid of oscillations, e.g. frontal cortex? Is it because of some specific projection patterns, or because of some micro-architectural features?

(R) We thank the reviewer for pointing out this facet of the data, which has intrigued us for a long time as well. There likely are multiple synaptic- and micro-circuit control parameters that affect the lifetime of oscillations, depending on the frequency and the generating mechanisms of said oscillations. For instance, activity in PFC is dependent on a precise neurochemical control noradrenaline and dopamine, or of acetylcholine (Cools & Arnsten 2022), which influence also oscillation dynamics as shown by pharmacological modulation (Van Der Brink et al., 2018) and genetic variability (Simola et al., 2022). At the macroscopic level, rhythmicity is also affected by endogenous and emergent critical dynamics, which vary greatly across brain regions and individuals (Fusca et al., 2023)..

As an intriguing dimension to this conundrum, there appears to be a high degree of correlation between the ‘oscillatoriness’ of cortical areas and their evolutionary stage, so that oscillations are strongest in the evolutionarily oldest areas

(sensorimotor) and weaker in the newer ones (association cortices).

Moreover, the anterior prefrontal cortex, which is thought to be largely specific to homo-sapiens and also evolutionarily the area with most recent expansion in humans, is largely devoid of oscillations.

We also used resting-state data for the oscillatory maps, It is possible that the prefrontal cortex shows intrinsic oscillatoriness only in the higher cognitive processing functions such as the working memory (Lundqvist & Miller, 2016).

(Action) We have now added these considerations to a new discussion paragraph where, in particular, we discuss the influence of neuromodulation “evolution of oscillations” and the resting-state hypothesis.

Section Discussion, lines 424-430:

Meanwhile, activity in PFC is dependent on a precise neurochemical control noradrenaline and dopamine, or of acetylcholine (Cools & Arnsten 2022), which influence also oscillation dynamics as shown by pharmacological modulation (Van Der Brink et al., 2018) and genetic variability (Simola et al., 2022). Moreover, the anterior prefrontal cortex, which is thought to be a largely homo-sapiens specific and evolutionarily the area with most recent expansion in humans and plays a role in higher cognitive functioning, was largely devoid of oscillations in our data. It is possible that PFC shows intrinsic oscillatoriness only in the higher cognitive processing functions such as the working memory while it is silent in resting state (Lundqvist & Miller, 2016).

(R, continued) We approached the influence of neuromodulatory system to oscillatoriness idea by testing whether the presence of oscillations in MEG data (measured with the fraction of significant pACF findings in any frequency across subjects per parcel, as in Fig.4) was correlated with the density of neurotransmitter receptors and transporters (Hansen et al., 2022). Interestingly, in high-alpha and beta bands, the rhythmicity was negatively correlated with most neuromodulatory systems save for the norepinephrine system (Figure for Reviewers, Fig. R1A). The strongest negative correlations were found with the mu-opioid receptors at low alpha (9.8Hz, $r^2 = -0.59$), low beta (19.7Hz, $r^2 = -0.65$) and serotonin receptors at 11.89Hz ($r^2 = -0.51$) that have a high density in frontal regions (Fig. R1B). Significant positive correlations were found with dopamin- and norepinephrinergic systems at low alpha (7.39Hz, $r^2 = 0.36$) and mid-beta (14.18Hz, $r^2 = 0.36$) while the frontal areas tend to have lower concentration of these receptors (Fig. R1B).

(A) We did not include the content in Fig. R1 in the present revision of the manuscript as we fear that it would overload the already fairly extensive study. We are preparing a separate publication about these findings. However, if Reviewer #1 feels that these data would considerably strengthen the present manuscript, we are ready to add these findings also here.

- Page 20. The work does not actually use “DTI”, I’d advise the authors to change the subsection title to “DWI”

(R) We thank the reviewer for pointing this out and apologise for the mistake.

(A) We have changed the subsection title.

Figure for Reviewers R1.

Rhythmicity across the neocortical mantle is associated with densities of neuromodulator receptors and transporters. A

Heatmap of the Pearson correlation coefficient between the cortical topographies of the fractions of significant oscillations and densities of neuromodulator receptors and transporters. On the right side the average fraction of positive (red) and negative (blue) correlations for each receptor. **B** Scatter plots of the fractions of significant oscillations vs. densities of D_2 , Mu , $5-HT_{2A}$ and NET receptors. The anatomical maps indicate the corresponding receptor density profiles.

Reviewer #2 (Remarks to the Author):

In the submitted manuscript, a new index for quantifying the level of rhythmicity of electrophysiological brain signals is introduced and applied to MEG and intracranial EEG recordings of human participants. This index is derived from the autocorrelation function of the instantaneous phases of the signal and is referred to as the phase autocorrelation function (pACF) in the manuscript. The index itself is referred to as the lifetime of the phase autocorrelations. Although this is an interesting approach, there are several methodological issues, unaddressed control analyses, and the claims are not completely corroborated by the analyses results (details below).

The manuscript is well-structured but poorly written: At many places in the text, it is difficult to find out exactly what was done. Furthermore, several phrases and statements are unclear, unnuanced, or simply incorrect. Some examples are given in the minor comments below, but there are several more that I didn't list.

The method is not well worked out theoretically and there seems to be a fundamental problem with hypothesis testing for significant rhythmicity based on the proposed index (see major comment 2). Specifically, the null hypothesis is left implicit and when made explicit, it becomes clear that it does not coincide with the desired null hypothesis (the absence of rhythmicity) but with the absence of (ordinary) autocorrelations.

Reply (R) We thank the reviewer for their effort put into this review and for the rigorous and critical approach. We feel that addressing these comments has significantly strengthened the manuscript.

Moreover, we would like to clarify our position with respect to the fundamental problem proposed by the reviewer here: We cannot see that there could be "rhythmicity" in an absence of autocorrelations because oscillations by definition are a process involving autocorrelations. For example, at the level of observables, oscillations are historically operationalized with an autocorrelation function that exhibits at least one side-peak at the characteristic time scale.

Action (A) As detailed below, we have now included new control analyses and corroborated the claims with both theoretical and empirical arguments, as detailed below. We have also worked extensively throughout the manuscript to improve the writing so that it is more clear and accurate. Finally, we have clarified both in the presented results and replies here to address the concerns about hypothesis testing.

Pertaining to the fundamental problem above, we clarify the relationship between the autocorrelation function and phase-autocorrelation function now in detail in the replies and actions to Comment 2.

MAJOR COMMENTS

1[A]. The advantage for quantifying the level of rhythmicity of a signal in terms of the pAFC instead of the power spectrum is poorly explained. In the Introduction (Line 53) the authors write

“However, being amplitude-based, this approach is only a qualitative indicator of rhythmicity and does not quantify it explicitly”.

First, being amplitude-based and being qualitative are independent issues. Amplitude-based measures of rhythmicity can be both qualitative and quantitative and this holds for any other measure, whether based on amplitudes or not.

(R) We thank the reviewer for this comment, this is an important aspect to clarify. Our intention here was to indicate that power-based methods, by definition, cannot measure rhythmicity explicitly, with “explicitly” meaning, for example, an output that has “rhythmicity” on y-axis. More specifically, rhythmicity here is a construct that we operationalize with phase autocorrelations, and we claim that while this is an explicit and quantitative operationalization, such an inference cannot be achieved with amplitude-based metrics.

With “qualitatively”, we mean that while, e.g. the width of a power spectral peak is dependent on rhythmicity, and even if this width can be quantified, several confounders limit the interpretability and preclude the conclusion that a change in said width could be quantitatively associated with a change in rhythmicity. This is because rhythmicity is by definition a property contained in the temporal stability of the phase of a complex signal, and it is independent of its amplitude, which we demonstrate with simulated data in Fig. 1.

It is important to note that the PSD does not convey physiologically readily interpretable or comparable information about the nature of oscillations. For example, a peak in the power spectrum shows that oscillations exist in the data, but how rhythmic they are, is impossible to deduce as the PSD peak magnitudes are dependent on the signal power and there is no way to translate the PSD peak width into an unambiguous estimate of oscillation lifetimes.

In contrast, the pACF lifetimes do provide such an estimate and are universally comparable across all signal modalities(say SEEG and MEG).

(A) We made the next changes to the Manuscript:

Introduction, lines 55-63

However, being amplitude-based, this approach is only a qualitative indicator of rhythmicity and does not measure it directly nor quantify it explicitly. For example, a peak in the power spectrum shows that oscillations exist in the data, but how rhythmic they are, is impossible to deduce as the PSD peak magnitudes are dependent on the signal per se and there is no way to translate the PSD peak width into an unambiguous estimate of oscillation lifetimes.

Even though the width of the PSD peak is dependent on rhythmicity and can be quantified, several confounders limit the interpretability and preclude the conclusion

that a change in said width could be quantitatively associated with a change in rhythmicity. This is because rhythmicity is by definition a property contained in the temporal stability of the phase of a complex signal, and it is independent of its amplitude (Figure 1e,j)

1.[B] Second, it is unclear what the authors mean with the phrase “quantify it explicitly”. What does it mean to quantify something implicitly? The authors go on to write:

“For example, brief high-amplitude bursts have been shown to be a robust neuronal phenomenon and to play a role in cognitive functions but can not be dissociated from sustained and temporally sustained oscillations with power-spectral amplitude-based methods.”

The claim here is that amplitude-based methods cannot distinguish between transients and sustained oscillatory dynamics, but the exact same thing can be said about the pACF. Both methods are based on averaging over windows during which the signal is assumed to be stationary. Furthermore, the issue of non-stationarity is entirely unrelated to whether a measure is qualitative or quantitative and hence it is unclear why it is used as an example in this context.

(R) With all due respect, we would like to note here that pACF does depend on signal amplitude and therefore not incorporate such a stationarity assumption and will be appropriately estimated even for non-stationary data. To corroborate this statement, we computed pACF lifetime spectra for a stationary signal with an oscillatory component at 10Hz and the same kind of signal but varying its mean (non-stationary mean), standard deviation (non-stationary std) or both over time. pACF spectrum has the same shape, magnitude and peak for all those kinds of signals (see Figure for Reviewers R2).

Figure for Reviewers R2. pACF spectra are insensitive to non-stationary data. pACF lifetime spectra for stationary signal with oscillatory component, for the same type of signal but with non-stationary mean, standard deviation and combined mean with deviation.

(A) We incorporated the figure for Reviewers R2 to the new Supplementary Figure 1g in the manuscript and made the next changes to the text:

Methods, lines 619-621

An interesting feature of pACF, stemming from the fact that it uses only phase information, is that it does not assume that a signal is stationary. Indeed, analyses of simulated data corroborate this and show that the pACF spectra are the same for stationary and non-stationary signals (Supplementary Figure 1g).

I do understand that if one is interested in phase-dynamics of the signal, it is more natural to consider the pACF than the power spectrum, because the pACF is defined solely in terms of phases and doesn't refer to amplitudes (but see major comment 2).

(R) We thank the reviewer #2 for this constructive and insightful comment. We agree that pACF lifetime *per se* does not distinguish between transient bursts and rhythmic activity. However, this information can be extracted from the shape of the phase autocorrelation function, which we show in Figure 7a,d. Here, the shape of the pACF curve is used to compute a stability index that yields a direct and exclusively phase-based estimate of "burstiness". This index allows to dissect between stable activity even with low pACF lifetime and burst-like activity

(A) We have now clarified this notion which now reads as

Section Results, lines 262-268

Both animal-model (Sherman et.al. 2016) and human electrophysiological studies (van Ede et. al. 2018) suggest that the rhythmicity of neuronal oscillations may vary qualitatively from continuous 'meta-stable' oscillations to apparently discrete bursts lasting only a few cycles. Brief high-amplitude bursts (Little et.al 2019, Wessel et.al 2019) have been shown to be a robust neuronal phenomenon and to play a role in cognitive functions (Lundqist & Miller, 2016), but can not be dissociated from sustained and temporally stable oscillations with power-spectral methods.

Bursts are typically assessed using the amplitude envelope so that bursts are defined as periods with amplitude exceeding a given threshold but there is ongoing debate about how to characterise the burstiness of a signal as a whole.

as well as Section Discussion, lines 451-461

Recent primate electrophysiology research has shown that working memory demands might be dependent on burst-like activity rather than stable oscillations in the prefrontal cortex (Lundqist & Miller, 2016, Miller et.al. 2018). It also has been proposed that brain dynamics is composed from transient bursts rather than stable oscillations (van Ede et.al 2016) . Such bursts are typically defined as short periods of high-amplitude narrow-band activity (Cole et.al. 2019).

Methodological approaches to quantifying bursting activity and measuring burstiness have, however, remained an area of active development. Power-spectral methods cannot dissect stable and burst-like activity with a poor signal-to-noise ratio, hence the solutions have remained predominantly based on amplitude time-series analyses. These approaches match the visual intuition of bursts often exceeding the 'background activity' in amplitude, but leave implicit the notion of bursts possibly also pertaining to the temporal stability of neuronal oscillations.

2[A]. In the manuscript, “rhythmicity” is defined implicitly by the way the null-data is constructed. When it is made explicit, it becomes clear that “rhythmicity” is the same as “having (ordinary) autocorrelations”.

(R) This is a matter we already touched upon at the onset of these responses. Let us continue from there in additional detail. We would like to dissect the considerations of null data and the concept of (phase) autocorrelations. For null data, the challenges are dominated by filter-induced autocorrelations, which dictate the surrogate data generation demands - please see 2[B] for this aspect.

With respect to the concept of autocorrelations, we do agree with the notion that phase autocorrelations are related to “ordinary” autocorrelations. The main difference between these two metrics is that ordinary autocorrelations are based on the real-valued signal and therefore the amplitude of each sample biases the weight of said sample in the autocorrelation function. For this reason, the ordinary autocorrelation function retains amplitude as a confounder that could bias the estimates of oscillation lifetimes, which is a problem that is greatly mitigated by pACF. In analyses of simulated and real data, it is easy to find that this amplitude-bias leads to inflated autocorrelations in the ordinary autocorrelation function (please see Fig. R3) while an amplitude-normalized autocorrelation function actually co-localizes with the phase-autocorrelation function.

(A) This relationship between ordinary autocorrelations and phase autocorrelations has now been illustrated in both the Figure for Reviewers 3 and the new supplementary Figure 1 panels h,i.

Figure for reviewers R3. The classical autocorrelation function (ACF) yields inflated correlation values (Rho) driven by the autocorrelations of large-amplitude samples. (A) Autocorrelation functions (blue lines) of filtered pink noise and (B) of representative MEG parcel signal in the alpha-frequency band (9.53Hz). The ACFs (dashed blue lines) were computed using the real part of wavelet-transformed simulated (A) and real (B) data, and their envelopes were then obtained by using the Hilbert transform (solid blue lines). As a contrast, we performed the ACF analyses also on amplitude-normalized real parts of wavelet-transformed data (green lines). The decay of amplitude-normalized ACF matches very well the decay of pACF (red lines). Inflation of ACF (blue) by large-amplitude oscillations is salient in comparison with the amplitude-normalized (green) and phase autocorrelations (red).

(A) We have now also added a section to Discussion about the equivalence and differences between phase- and ordinary-autocorrelation functions.

Section discussion, lines 337-343

This approach is similar to the classical autocorrelation function that has been extensively used to quantify temporal stability in both broad- and narrow-band signals. The autocorrelation function, however, is based on real-valued signals and therefore the amplitude of each sample biases the weight of said sample in the autocorrelation function. For this reason, the ordinary autocorrelation function retains amplitude as a confounder that biases the estimates of oscillation lifetimes towards those of large-amplitude oscillations. This problem is greatly mitigated by pACF using only phase information and being independent of the amplitude (See Suppl. Fig. 1h,i).

2[B]. The null-data used by the authors is white noise, which is characterised by its autocorrelation function being a delta function at lag zero and this is equivalent to its power spectrum being flat. Hence the null-hypothesis is that, in the considered frequency range, the power spectrum is flat. This shows that the null-hypothesis cannot be expressed solely in terms of phases and that amplitudes need to be taken into account. More importantly, if the null-hypothesis is rejected, one can draw the conclusion that the power spectrum is not flat, i.e., the observed signal is not white-noise. This conclusion is very different from the desired conclusion which is the presence of rhythmicity.

(R) First, we must apologize for the confusion in our Methods section. We do not use white-noise in this study, in all analyses of real MEG and SEEG, the null data we have used has been $1/f$ noise. As detailed below, this makes no difference in practice but we wanted to clarify this writing error in the original manuscript.

(A) The error has now been corrected and the revised Method section reads:

We wavelet-filtered 10.000 realizations of the pink-noise data using the same wavelet parameters and matching the data length, applied the pACF lifetime pipeline and computed the 99th percentiles of the pACF lifetime values for each frequency

(R2) We agree that when the samples are independent, the autocorrelation will take the form of a delta function, e.g. in the case of white noise data. However, in the filtered narrow-band data, samples are never independent because of the filter-induced autocorrelations and the strength of such autocorrelations is a function of filter properties (in particular, number of cycles in the Morlet wavelet, supplementary Figure 1c). The filter-caused short-range autocorrelations are much stronger than those possibly present in the broad-band signal.

To corroborate this notion, we reproduced the lifetime estimates for different power-spectral scaling exponents for noise. The simulations showed that the null-level of the pACF lifetime was, indeed, very little affected by noise color and is a

flat line for both white noise and pink noise (see Figure for Reviewers R4).

Figure for reviewers R4. pACF spectra are independent of the noise colour. Power-spectrum (A) and pACF-lifetime spectrum (B) of noise signals generated with exponents varying from 0 (white-noise) to 1 (pink-noise). Thick lines indicate mean value, shaded areas indicate CIs estimated with standard deviation.

(A2) We have now clarified this notion and it now reads as
Section Methods, lines 604-606

However, narrow-band filtering induces artificial phase-autocorrelations into a data that are dictated by the time domain width of the filter. In the case of Morlet wavelets, the strength of these autocorrelations depend on the number of cycles in the wavelet (See Supplementary Figure 1c).

(A2 cont.) We also extended the Supplementary Figure 1 and added the two panels from Fig. R4 there.

(A2 cont.) We now also acknowledge in the Methods that even though the effect of the noise scaling exponent of pACF lifetime estimates is negligible, it would be best to estimate the null hypothesis lifetime distributions by using noise that has a scaling exponent matching that of the real data.

Section Methods, lines 692-696 now reads as

To test whether a pACF lifetime was significantly different from that of a random process, it is essential to take into account the filter-induced autocorrelations and to match the real-data scaling exponents (Supplementary Figure 1e,f). We wavelet-filtered 10,000 realizations of the pink-noise data using the same wavelet parameters and matching the data length, applied the pACF lifetime pipeline and computed the 99th percentiles of the pACF lifetime values for each frequency

3. The authors mention that most studies on electrophysiological signals use spectral instead of phase-based measures to characterise rhythmicity. The reason for this, I think, is that electrophysiological signals are nearly Gaussian and all information about a Gaussian signal is contained in its power spectrum.

(R) Prior to the discussion on pACF, we would like to note here that we respectfully disagree with some aspects of these premises.

- First, neuronal oscillations appear to be fairly strongly non-Gaussian, which is evidenced by their heavy-tailed (gamma-like) amplitude distributions rather than them exhibiting Rayleigh distributed amplitudes expected for Gaussian processes. Neuronal oscillation amplitude envelopes also exhibit power-law long-range temporal correlations in their power spectra, autocorrelation functions and detrended fluctuation analyses, which is incompatible with the notion of them being Gaussian.
- Second, while technically speaking “all information” in the signal is contained in both the original real-valued signal and its Fourier transform, this does not indicate that all information would be accessible in either of these data representations. The objective of pACF is to make phase autocorrelations explicitly accessible so that they can be used to operationalize rhythmicity.

3. (cont.) In particular, the pACF can be expressed in terms of the power spectrum and hence does not add information, unless the signal is non-Gaussian. I think this point should be made explicit early in the manuscript.

(R) Continuing from above, we see it essential to base the definition of pACF on a time-frequency decomposition of the signal so that it does not require the stationarity assumption intrinsic to power-spectral methods or lose information attributable to non-Gaussian aspects of neuronal signals.

It is also important to note that there already is a precedent to power-spectrum-based autocorrelation estimation, the ‘lagged-coherence’ method, which we cite in both the Introduction and Discussion. This method is interesting but also fundamentally and irreparably flawed by the breaking of the aforementioned assumptions. E.g., as we detailed already in the original submission, lagged-coherence yields incorrect peak frequencies for oscillations because of the bias caused by instantaneous-frequency dynamics of oscillatory processes (please see Section Methods, Phase autocorrelation estimates, lines 608-611 and Suppl. Fig. 1a for details)

(A) As suggested by Reviewer #2, we have now added a following note on this background into Introduction (page 2):

Section Introduction, lines 64-75

In addition, depending on the nature of the underlying dynamics, neuronal oscillations exhibit a wide range of amplitude distributions (Freyer et.al. 2012) ranging from Gaussian, to non-normal, to bistable (Wang et.al. 2023), which is evidenced by their heavy-tailed (gamma-like) amplitude distributions (Roberts et.al. 2015). If neuronal oscillations were Gaussian processes, they should exhibit Rayleigh distributed

amplitudes that have an exponential decay of autocorrelations, which has been disproved in a large body of empirical data. Oscillation amplitude envelopes also exhibit power-law long-range temporal correlations in their power spectra, autocorrelation functions and detrended fluctuation analyses in the neural data (Linkenkaer-Hansen et.al. 2011) and computational models (Cowan et.al. 2016), which is incompatible with the notion of them being Gaussian. All of those require development of approaches that do not make assumptions about underlying distribution of signal observables.

Here we advance a new approach and operationalize the construct of rhythmicity with the phase autocorrelation function (pACF). It specifically quantifies the rhythmicity of neuronal oscillations in an amplitude-independent manner and expresses the predictability of a future phase as a function of time (lag), thereby yielding a direct measurement of temporal stability, i.e., rhythmicity.

3. (cont.) To demonstrate that the pACF actually adds information, one has to show that some aspect of the pACF (in this case the lifetime) is unlikely to be drawn from the distribution of lifetimes under the Gaussian null-hypothesis (as quantified by a p-value). The null-distribution of lifetimes can be approximated by calculating the lifetimes of a large number of phase-randomised copies of the observed signal.

(R) We thank Reviewer #2 for pointing this out. Nonetheless, as described in the previous comment (please see major comment 2[B] above), the phases in the null-data are not independent due to filter-induced autocorrelations, therefore randomising phases of the null-data signals will negate it and lead to underestimated baseline level (see Figure for Reviewers R5).

Figure for reviewers R5. Wavelet-filtering induces spurious phase autocorrelations into data. pACF lifetime as a function of frequency for filtered noise, for data with a simulated oscillatory component at 10Hz, and for phase-shuffled data with an oscillatory component.

As mentioned in the Methods section, to assess significance of pACF lifetime, we adapted the non-parametric approach and compared real-data lifetimes with the 99th percentile of the null-level values which correspond to p-value < 0.01.

MINOR COMMENTS

1. At line 301 the authors write

“Neuronal oscillations encompass a wide range of neuronal processes where the phase of the oscillation is the key mechanistically and functionally significant element.”

This is a significant statement that should at least be backed up by references. It is also true that amplitude fluctuations play an important role in brain function, given that the resting-state networks reflect correlations between amplitudes and not phases. I therefore think that the above statement should be nuanced.

(R) We apologise for not comprehensively citing the prior art. We have now added several citations to this work about the role of phases in brain function. We would also like to point out that the resting-state networks also encompass extensive phase correlations. The relationship of phase and amplitude correlations as two operationalizations of ‘functional connectivity’ has remained an active topic for investigation.

(A) We have now extended the description of the role of the phases and added the next changes:

Discussion, lines 323-328

Neuronal oscillations encompass a wide range of neuronal processes where the phase of the oscillation is the key mechanistically and functionally significant element (Buzski et.al 2006, Palva & Palva 2012, Thut 2012), they predict visual perception (Busch et.al. 2009), plays a role in neural coordination (Volloh et.al. 2016), plasticity (AndradeTalavera etl.al .2023) and reflect phase-specific functional relations between neuronal populations (Hindriks et.al. 2023). Phase-synchronisation is also a basis for resting-state networks (Williams 2023, Fusca 2023, Brookes 2011), relations between phase and amplitude correlations (Zhigalov 2017, Arnulfo 2015) and between different frequencies (Siebenhner 2020) has remain an active topic for investigation.

2. It will be good if the authors distinguish between the population and sample pACF, at least in the methods section. Also, when defining the pACF in the methods section, the authors might want to add that the lifetime is zero if and only if the instantaneous phases at different time-points are independent (the pACF will then be a delta function).

(R) We thank the reviewer for this note. We added the necessary clarification to the text.

(A) We made the next changes to the text:
Methods section, lines 593-601

To estimate the phase autocorrelation function of an analytical signal (e.g. after wavelet-filtering), we calculated the Phase Locking Value (PLV) of a signal with a delayed

copy of itself. The PLV is the absolute value of the complex PLV (cPLV)(Palva et.al., 2012), population pACF defined as

$$pACF(l) = E[CS_{x,x(l)}]$$

Where E[.] denotes the expected value and $CS_{x,x(l)}$ denotes cross-spectrum between a complex signal X and a copy delayed with lag l. In practice, pACF is estimated using limited data and sample pACF is defined as

$$\overline{pACF(l)}_{sample} = \frac{1}{N} \sum_{i=1}^N CS_{x,x(l)}$$

where N is the total number of samples.

Methods section, lines 603-604

Note that pACF will be a delta function if signal phases are independent and a constant if signal phases are linearly dependent.

3. At time 321 the authors write

“Explicit measurements of rhythmicity used to be, in fact, commonplace in early years of research on neuronal oscillations when both autocorrelation functions of continuous data and histograms of spiking were routinely used to quantify rhythmicity. Recently, however, the methodological focus has turned to power spectral approaches...”

First, power spectral methods are not recent but are, in fact, the classical way to analyze (neuronal) signals. Second, being Fourier transforms of each other, the autocorrelation function and the power spectrum provide the same information. It is therefore puzzling to me why the authors contrast the two.

(R) We thank the reviewer for pointing towards this inconsistency. We fully agree that power-spectral methods are classical approaches as well. Our intention was to convey that still in the 90s it was commonplace to use autocorrelation functions and histograms to quantify the oscillatoriness *per se* in neuronal signals. Power spectra were used back then also to indicate the presence of oscillations, even though they do not give a direct read-out of the lifetime of oscillations like the autocorrelation function does. I.e., we mean that the usage of ACFs has been declining while PSD-based methods have remained an active topic of development.

(A) We made the next changes to the text:

Discussion section, lines 354-359:

Indeed, explicit measurements of rhythmicity were commonplace in the early years of neural oscillations research (Singer et.al.2018, Gray et.al. 1989), when both autocorrelation functions of continuous data (Stenkamp et.al. 2001, Brazier and Barlow, 1956) and histograms of spikes (Murean et.al. 2008) were routinely used to quantify rhythmicity, while the power spectrum was used to detect the presence of oscillations. Over time, however, the use of classical autocorrelation-based methods has declined

and the power-spectrum approaches have become the primary way of analysing oscillations in general, where oscillations are defined relationally as frequencies with greater power than neighbouring frequencies.

4. At line 325 the authors write

“We advance here a new approach to quantify rhythmicity explicitly and objectively with phase autocorrelations.”

This gives the impression that spectral methods are, somehow, not objective. Please phrase this more carefully and accurately.

(R) The sentence reads about quantification of rhythmicity, not about quantification overall. pACF quantifies rhythmicity objectively while spectral methods do not, even though some aspects of spectral features (like peak width) may be proportional to rhythmicity.

(A) The manuscript now reads as:

Section Discussion, lines 360-363

In this approach, oscillatory power is qualified as a function of frequency and the concept of rhythmicity can only be indirectly operationalised as the width of the spectral peak, which can still be confounded by other signal properties. Here we advance a new approach to quantify rhythmicity explicitly and objectively using phase autocorrelations.

5. The authors define the “lifetime” of the phase autocorrelations and note that this is “similar to how exponential decay is indexed by a decay time constant”. However, for the exponential decay, there is a natural time-constant, whereas this is not the case for the pACF. It will be good to mention this.

(R) We thank the reviewer #2 for pointing out the difference with the exponential decay. We rewrote the sentence.

(A) We made the next changes to the Manuscript:

Section Results, lines 84-87

The pACF can then be aggregated into a single value to indicate the "lifetime" of phase autocorrelations similar to how exponential decay is indexed by a decay time constant. However, as we make no assumptions about the shape of the pACF curve, we have used a model-free approach by testing where the CDF crosses a threshold.

6. The distribution of the null-data is not mentioned. In any case, it will be different from that of the observed signals, which is undesirable, because rejection of the null-hypothesis can then be due to the observed signal having a different distribution than the null-data and not due to the presence of rhythmicity.

(R) We are not quite sure which distribution of the null-data Reviewer #2 means here. I.e., is this question about the amplitude distribution of the surrogate data, phase-autocorrelation values of surrogate pACF, or about the surrogate pACF lifetimes? To address this case we directly compared distributions of the null-level pACF lifetimes and the pACF lifetimes obtained from the real data. We found that these are similar (Kolmogorov-Smirnov statistic = 0.39, p-value < 0.001), except for the heavy tail of significant lifetimes in the real data.

Figure for reviewers R6. Distribution of the filtered-noise pACF lifetimes is similar to the real-data. Distribution of pACF lifetime of 500 realisations of the filtered pink-noise data (black), 99 percentile of this distribution (black dashed line) and distribution of pACF lifetimes at frequency of 8.9Hz for a single SEEG subject (blue). The oscillatory activity is characterised by prolonged oscillations with lifetime higher than noise-level.

(A) We have now added a comparison of the pACF lifetime distributions to the manuscript. We have also incorporated the Figure for Reviewers R6 into the Supplementary Figure 2 as a new panel d. The manuscript additionally reads:

Section Results, lines 110-111

This showed that while distribution of pACF lifetime values for contacts without oscillations was similar to the distribution of noise-level values (Supplementary Figure 2d) a subset of brain areas exhibited significant oscillations (Figure 2c).

7. On line 327 the authors mention that quantifying rhythmicity as done in ref. [56] has a

“frequency-shifting bias that leads to erroneous peak-frequency estimates”

and then refer to Supplementary Figure 1a. However, this figure is difficult to understand and so it remains unclear what the above actually means. For instance, the legend of panel a reads “Morlet wavelets ...” but no wavelets are shown and it is unclear how it relates to wavelets. Please explain the above bias-issue and make the supplementary information self-contained.

(R) We apologise that the description was unclear in the previous version. We made appropriate changes to the Methods section and to the figure caption to make the text more clear.

(A) To ensure that the supplementary information is self contained, we have now made the following changes to the manuscript:

Methods section, lines 608-614

In the regular way, the transformation of lag in cycles to samples would be done by multiplying it on the sampling frequency and dividing on the frequency of interest $lag * \frac{SRate}{Frequency}$. However, this approach leads to a biased pACF spectral peak towards higher frequencies; higher than the frequency of the ground truth oscillation and the PSD peak (Supplementary Figure 1a).

In order to overcome this issue, we applied an instantaneous frequency correction where we divided the sampling frequency on the mean instantaneous frequency of the wavelet-filtered data rather than its central frequency $lag * \frac{SRate}{\overline{IF}}$ where \overline{IF} is a mean instantaneous frequency

Supplementary Figure 1a legend

PSD and pACF lifetime values of a signal with a simulated oscillatory component at 10Hz computed with and without instantaneous-frequency correction. pACF lifetime estimated without the instantaneous-frequency correction term (see the black dashed line) is biased towards higher frequencies and its peak is located incorrectly at 12Hz. However, the pACF computed with instantaneous-frequency correction (see the red dashed line) exhibits the peak correctly at 10 Hz, as verified by the PSD spectrum.

8. On line 330 the authors write

“Using simulations, we established here that rhythmicity and amplitude, as operationalized with pACF lifetime and signal power, are theoretically and analytically double-dissociable.”

I’m not sure what this sentence means. Are the authors referring to the fact that the pACF is independent of the power of the signal? Furthermore, what does “analytically” mean here and how is it different from “theoretically”?

(R) We apologise that the description was unclear in the previous version. With “analytically” we meant experimentally (including using both simulated and real data). (A) We have now clarified the text as follows:

Discussion section, lines 366-368

Using simulations and real MEG and SEEG recordings, we established here that rhythmicity operationalized with pACF lifetime and amplitude operationalized with signal power tap to different constructs and are theoretically and experimentally double-dissociable.

REVIEWERS' COMMENTS:

Reviewer #1 (Remarks to the Author):

The authors have comprehensively addressed my concerns and I recommend publication.

Reviewer #2 (Remarks to the Author):

The authors have made a substantial effort to address the raised issues and I believe they now have made a strong case for their claims.